# The Structural Complexity of Matrix-Vector Multiplication

**Emile Anand**[*]
Georgia Institute of Technology
Atlanta, GA 30308
emile@gatech.edu

**Jan van den Brand**
Georgia Institute of Technology
Atlanta, GA 30308
vdbrand@gatech.edu

**Rose McCarty**[†]
Georgia Institute of Technology
Atlanta, GA 30308
rmccarty3@gatech.edu

## Abstract

We consider the problem of preprocessing an $n \times n$ matrix $\mathbf{M}$, and supporting queries that, for any vector $v$, returns the matrix-vector product $\mathbf{M}v$. This problem has been extensively studied in both theory and practice: on one side, practitioners have developed algorithms that are highly efficient in practice, whereas on the other side, theoreticians have proven that the problem cannot be solved faster than naive multiplication in the worst-case. This lower bound holds even in the average-case, implying that existing average-case analyses cannot explain this gap between theory and practice. Hence, we study the problem for *structured* matrices. We show that for $n \times n$ Boolean matrices of VC-dimension $d$, the matrix-vector multiplication problem can be solved with $\widetilde{O}(n^2)$ preprocessing and $\widetilde{O}(n^{2-1/d})$ query time. Given the low constant VC-dimensions observed in most real-world data, our results posit an explanation for why the problem can be solved so much faster in practice. Furthermore, we show how to extend this result to the non-Boolean setting with the Pollard pseudodimension.

Our results yield the first non-trivial upper bounds for many applications. In previous works, the online matrix-vector (OMv) hypothesis (conjecturing that quadratic time is needed per query, even over the boolean semi-ring) was used to prove many conditional lower bounds, showing that it is impossible to compute and maintain high-accuracy estimates for effective resistance, Laplacian solvers, shortest paths, and triangle detection in graphs subject to node insertions and deletions in subquadratic time. Yet, via a reduction to our matrix-vector-multiplication result, we show we can maintain these problems efficiently if the input is structured, providing the first subquadratic upper bounds in the high-accuracy regime.

## 1  Introduction

Computing sequential matrix-vector products is a fundamental subroutine of many iterative algorithms in machine learning. In optimization [Renegar, 1996, Vempala et al., 2020, Lin et al., 2023a, Chaudhari et al., 2024], computational geometry [Har-Peled, 2009, Welzl, 1992, Fisikopoulos and Penaranda, 2016], online algorithms [Lin et al., 2024, Anand and Qu, 2024, Murray et al.,

---

[*]Work partially done while at Cognition AI
[†]Work done while at Princeton University.

39th Conference on Neural Information Processing Systems (NeurIPS 2025).

2021], and dynamic algorithms [Liu, 2024, Jiang et al., 2023, Anand et al., 2024b, Jin and Xu, 2022], sequential matrix-vector products are essential subroutines underlying performant algorithms. The current learning revolution is powered by hardware specifically designed to perform such products, as both neural network evaluation and back-propagation require repeated matrix-vector products [Sørensen, 2012, Rumelhart et al., 1986]. Hence, matrix-vector products are arguably one of the most important and fundamental subroutines, with any complexity improvement having wide-ranging implications, and is thus a prevalent research topic in both theory and practice.

The problem can be modeled via the following data structure task: construct a data structure that preprocesses a given $n \times n$ matrix $\mathbf{M}$. After preprocessing, we want to multiply vectors with $\mathbf{M}$ faster than naive matrix-vector multiplication. Depending on the application, this matrix $\mathbf{M}$ is fixed or the data structure may need to handle changes to $\mathbf{M}$. For instance, when calculating probabilities of a random walk (e.g., in the PageRank algorithm [Page et al., 1999]), performing the power-method, and evaluating a neural network, the matrix is fixed. Conversely, in convex optimization, such as when solving a linear or semi-definite program, applying Newton-Raphson's method, and training a neural network, the matrix is the inverse of a Hessian or is given by the network's weights, which changes from one iteration to the next [Arriaga and Vempala, 2006, Anand and Liaw, 2025].

Beyond speedups due to improved hardware, practitioners have made tremendous progress in accelerating the computation of matrix-vector products through heuristics that run in $\mathrm{nnz}(\mathbf{M})$ (the number of non-zero entries of $\mathbf{M}$) worst-case time, but which are much faster in practice. This has led to speedups in training GNNs [Alves et al., 2024] and other algorithms [Floros et al., 2024].

Conversely, from a theory perspective there are substantial lower bounds. Gronlund and Larsen [2015] showed that any $\mathrm{poly}(n)$-space data structure for matrix-vector multiplication over sufficiently large fields (for instance, $\mathbb{R}$) require $\Omega(n^2/\log n)$ time. This quadratic lower bound was also proven for arithmetic circuits [Frandsen et al., 2001]. While these are worst-case lower bounds, they also hold for the average case [Henzinger et al., 2022]. The only non-trivial upper bounds beating quadratic time are over the Boolean semi-ring (i.e., $\{0, 1\}$ with $x \oplus y = \min(1, x + y)$). In a line of works that reduced the problem to finding neighborhoods in a graph and smaller algebraic products, this query complexity was improved to $O(n^2/\log^2 n)$ [Liberty and Zucker, 2006, Williams, 2007] and $O(n^{2-o(1)})$ [Larsen and Williams, 2017, Chakraborty et al., 2018, Abboud et al., 2024]. However, none of these algorithms are truly subquadratic, i.e., $O(n^{2-\epsilon})$ for some constant $\epsilon > 0$.

Moreover, while the $\Omega(n^2)$ lower bounds only hold over finite fields, even over the Boolean semi-ring it is conjectured that no truly subquadratic time algorithm exists [Henzinger et al., 2015], and this is commonly referred to as the OMv (online matrix-vector multiplication) conjecture. In fact, this has led to a recent line of works that use the conditional hardness of OMv to prove tight time lower bounds for many dynamic algorithms, such as dynamic matrix inversion [Brand et al., 2019a,b], dynamic subgraph connectivity [Henzinger and Neumann, 2016], dynamic regression [Jiang et al., 2023], dynamic range, Langerman's problem [Jin and Xu, 2022], and the generalized Klee's high-dimensional measure problem [Yildiz and Suri, 2012].

In summary, there is a notable difference between the perspectives of this problem in theory and practice: there are heuristics that are fast in practice, but from a theoretical perspective the problem is hard, and an entire subarea of fine-grained-complexity has been built on this hardness assumption [Hu and Polak, 2024]. Further, average-case analyses cannot explain this gap, as even the average-case is provably hard. In this work, we resolve this conflict: since the average-case (i.e., random non-structured inputs) is hard, the observed efficiency of practical algorithms must stem from some *inherent structure* in real-world data. Thus, we study the complexity for structured inputs.

A popular measure for structural complexity is the Vapnik-Chervonenkis (VC) dimension, which finds many applications in machine learning [Shalev-Shwartz and Ben-David, 2014, Bartlett et al., 2019], structural graph theory [Nguyen et al., 2024, Alon et al., 2016, Karczmarz and Zheng, 2024], and computational geometry [Har-Peled, 2009, B. Chazelle, 1989, Fisikopoulos and Penaranda, 2016] (see the preliminaries in Section 2 for a definition of the VC-dimension). For Boolean matrices, the VC-dimension has the following, more intuitive, description: it is the size of the largest subset $S \subseteq [n]$ of columns such that the rows of the sub-matrix whose columns are restricted to those in $S$ contain every possible string in $\{0, 1\}^{|S|}$. Many structured objects studied by theoreticians have inherently low VC dimension. For instance, star and interval graphs, and rank-one projection matrices, have VC-dimension 2, planar graphs have VC dimension 4, and any $H$-minor free graph has VC dimension at most $|V(H)| - 1$. Beyond being an established complexity measure in theory, it has also been

empirically observed that real-world data has low VC dimension [Coudert et al., 2024]. We explain this observation via the following structural characterization of matrices with low VC dimension:

**Theorem 1.1** (Proof in Appendix D). *Consider a hereditary class of $0/1$-matrices $\mathcal{M}$, meaning that $\mathcal{M}$ is closed under row/column deletion (i.e., each matrix $\mathbf{M} \in \mathcal{M}$ is still in $\mathcal{M}$ after deleting a row or column). If $\mathcal{M}$ is non-trivial, (i.e., does not contain every possible matrix), then there exists an absolute constant $c \in \mathbb{N}$ such that the VC-dimension of any matrix in $\mathcal{M}$ is at most $c$.*

Presumably, any unknown structure common to real-world matrices should not be lost when deleting rows/columns. Theorem 1.1 does not exclude other matrices (which do not satisfy this hereditary property) from having constant VC-dimension, e.g., adjacency matrices of minor-free graphs.

## 1.1 Our Results

While theoretical objects such as grid-graphs, intersection graphs, and kernel-matrices have a low VC-dimension, they are not perfect representations of real-world data. For instance, while city street networks are predominantly grid-graphs, there always exists exceptions in the form of bridges or few diagonal streets. Exceptions can also occur due to errors in measurements or other corruptions. To capture such *almost low VC-dimension objects*, we define the "corrupted VC-dimension."

**Definition 1.2** (Corrupted VC-dimension d). *The "corrupted VC-dimension" of a set system $\mathcal{F}$ with $m$ sets is the smallest $d$ such that there is another set system $\mathcal{F}'$ of VC-dimension $\leq d$ and $\mathcal{F}$ can be obtained from $\mathcal{F}'$ by adding and/or removing each element to/from at most $O(m^{1-1/d})$ sets.*

For matrices $\mathbf{M} \in \{0,1\}^{m \times n}$, a corrupted VC-dimension $d$ implies the existence of a matrix $\mathbf{L} \in \{0,1\}^{m \times n}$ of VC-dimension $\leq d$ and a 'corruption' matrix $\mathbf{S} \in \{-1,0,1\}^{m \times n}$ with at most $O(m^{1-1/d})$ non-zero entries per column, such that $\mathbf{M} = \mathbf{L} + \mathbf{S}$. Note that the mere existence of such an $\mathcal{F}'$ (i.e., $\mathbf{L}$) of VC-dimension $\leq d$ suffices for $\mathcal{F}$ (i.e., $\mathbf{M}$) to have corrupted VC-dimension $d$. We do *not* need to know $\mathcal{F}'$ or which sets have been corrupted. Also, the VC-dimension of $\mathcal{F}$ upper bounds the corrupted VC-dimension of $\mathcal{F}$. Hence, while we state our results for matrices with corrupted VC-dimension $d$, the proven upper bounds also hold for matrices with VC-dimension $d$.

We now state the theoretical guarantees of our accelerated matrix-vector multiplication algorithm in Theorem 1.3, where the runtimes are parameterized by the corrupted VC dimension measure.

**Theorem 1.3.** *(Static Online Matrix-Vector Multiplication). If a matrix $\mathbf{M} \in \{0,1\}^{m \times n}$ has corrupted VC-dimension $d$, then after an $\widetilde{O}(mn)$-time preprocessing, there is a data structure $\mathcal{D}$ that can compute $\mathbf{M}v$ for any $v \in \mathbb{R}^n$ in $\widetilde{O}(nm^{1-1/d} + m)$ time, with high probability.*

The problem of approximating the VC-dimension of any set system within a $(2 - \epsilon)$ accuracy for any $\epsilon > 0$ is known to be $\Sigma_3^P$-hard [Mossel and Umans, 2002]; however, for our theorem, the algorithm does not need to know or compute or even approximate the VC-dimension to achieve this runtime.

Our data structure is based on Björklund and Lingas [2001] which proposed an algorithm for Boolean matrix multiplication. Their algorithm computes the matrix-matrix product via $n$ matrix-vector products, thus implicitly providing a matrix-vector multiplication data structure as well. Their complexity depends on the weight of the minimum spanning tree (MST) defined with respect to the Hamming-distance between the rows of $\mathbf{M}$. However, the weight of the MST is $O(n^2)$ in the worst-case.

This algorithm was recently independently rediscovered in the graph neural networks community [Alves et al., 2024] which experimentally verified its efficiency on real-world inputs. Our Theorem 1.3 now gives a theoretical explanation as to why these practical algorithms are much more efficient in practice than the worst-case and average-case lower bounds of $\Omega(mn)$.

We obtain Theorem 1.3 via techniques from computational geometry. A line of works [Welzl, 1988, B. Chazelle, 1989, Matoušek, 1991, Har-Peled, 2009] consider the following *geometric intersection* data structure problem: given a set of points in $\mathbb{R}^d$, preprocess them. Then, given a convex polytope for each query, return whether any of the points intersect the polytope. However, prior work had exponential preprocessing time [Welzl, 1992] or unspecified polynomial time [Matoušek, 1991]. Matrix vector products can be interpreted as $O(m)$ intersection problems if we only care about Boolean outputs, because in the Boolean case $(\mathbf{M}v)_i \neq 0$ if and only if there is an intersection between the position of the non-zero elements in $v$ and the non-zero elements in the $i$-th row of $\mathbf{M}$. We extend these computational geometry techniques to non-Boolean outputs, and to our more general notion of corrupted VC-dimension. We provide details of this proof in Section 3.2.

In addition to our novel complexity bound, we also provide alternative algorithms to those presented in Björklund and Lingas [2001], Alves et al. [2024]. We provide more details below.

**Dynamic Setting.** We extend Theorem 1.3 by allowing $\mathbf{M}$ to undergo row and column updates. The proof is given in Appendix B.3.

**Theorem 1.4.** *(Dynamic Online Matrix-Vector Multiplication). Given a matrix $\mathbf{M} \in \{0,1\}^{m \times n}$, there is a data structure $\mathcal{D}$ with $\widetilde{O}(mn)$ preprocessing time that supports row and column updates (insertions/deletions) to $\mathbf{M}$ in $\widetilde{O}(n)$ and $\widetilde{O}(m)$ time, respectively. Upon querying $\mathcal{D}$ with a vector $v \in \mathbb{R}^n$, it outputs $\mathbf{M}v$ in $\widetilde{O}(nm^{1-1/d^*} + m)$ time, with high probability, where $d^*$ is the largest corrupted VC-dimension of $\mathbf{M}$ throughout the history of its updates.*

**Transposed Matrices.** If the VC-dimension of $\mathbf{M}^\top$ is $d$, then the VC-dimension of $\mathbf{M}$ is at most $2^{d+1}$ [Assouad, 1983], so Theorem 1.3 runs in $\widetilde{O}(nm^{1-1/2^d} + m)$ time. To address this slow-down, we extend the algorithm to bound its complexity by the VC-dimension of $\mathbf{M}^\top$.

If $\mathbf{M} \in \{0,1\}^{m \times n}$ and $\mathbf{M}^\top \in \{0,1\}^{n \times m}$ have corrupted VC-dimension $d$ and $d'$ (respectively), then after an $\widetilde{O}(mn)$-time preprocessing, we show we can compute $\mathbf{M}v$ in time $\widetilde{O}(\min\{nm^{1-1/d} + m, mn^{1-1/d'} + n\})$ with high probability. We do not need to know $d$ or $d'$, as the algorithm automatically achieves the minimum.

**Non-Boolean Matrices.** The Pollard pseudodimension extends the VC-dimension to non-binary thresholds. It finds many applications in theoretical machine learning [Shalev-Shwartz and Ben-David, 2014, Bartlett et al., 2019] and structural graph theory [Karczmarz and Zheng, 2024] (see Appendix B.4 for a definition of the *Pollard pseudodimension*). For $\mathbf{M} \in \mathbb{R}^{m \times n}$, if $d$ bounds the Pollard pseudodimension of $\mathbf{M}$ and if $T$ bounds the number of unique values per column of $\mathbf{M}$, then by Theorem B.9, $\mathbf{M}v$ can be computed in time $\widetilde{O}(Tnm^{1-1/d} + m)$. For $T \leq \widetilde{O}(1)$, this is subquadratic.

## 1.2 Applications

**Random Walks and PageRank by Scaling.** Let $\mathbf{A} \in \mathbb{R}^{n \times n}$ be the adjacency matrix of an $n$-node graph and let $\mathbf{D} \in \mathbb{R}^{n \times n}$ be its degree matrix. The normalized Laplacian is $\widetilde{\mathbf{L}} = \mathbf{I} - \mathbf{D}^{-1/2}\mathbf{A}\mathbf{D}^{-1/2}$. Even though $\widetilde{\mathbf{L}}$ is not Boolean, we can still use Theorem 1.3 since $\widetilde{\mathbf{L}}v = \mathbf{D}^{-1/2}(w - \mathbf{A}w)$ for $w = \mathbf{D}^{-1/2}v$ and the product $\mathbf{A}w$ can be computed via Theorem 1.3. The same technique extends to random-walk probability matrices. In turn, this allows fast simulations of Markov chains to compute stationary distributions and iterative computation of the PageRank [Page et al., 1999] via Theorem 1.3.

**Complexity-Theoretic Implications.** The Boolean matrix multiplication (BMM) conjecture states that no algorithm can multiply two boolean $n \times n$ matrices in $O(n^{3-\epsilon})$ time for constant $\epsilon > 0$ [Basch et al., 1995, Williams and Williams, 2010]. This conjecture is used to prove lower bounds on combinatorial algorithms (i.e., algorithms that do not make use of Strassen-style fast matrix multiplication methods), motivated by the property that *fast matrix multiplication*, while being theoretically fast (currently $O(n^{2.372})$ Williams et al. [2024]), is very slow in practice due to large constants hidden in $O$-notation. Hence BMM gives a lower bound on more practical algorithms and can be used to prove that any further complexity improvement would require fast matrix multiplication.

Theorem 1.3 implies that for matrices with corrupted VC-dimension $d$, BMM can be solved in $\widetilde{O}(n^{3-1/d})$ time, allowing for improved combinatorial upper bounds. The BMM problem was the original motivation for Björklund and Lingas [2001], Gasieniec and Lingas [2003]. Moreover, as demonstrated in Alves et al. [2024], this algorithm is also highly efficient in practice and does not encounter the large constant of fast matrix multiplication. Therefore, Theorem 1.3 implies that problems with lower bounds based on BMM can be solved more efficiently on structured inputs.

**Corollary 1.5.** *Suppose we are given matrices $\mathbf{M}_1, \mathbf{M}_2 \in \{0,1\}^{n \times n}$, where $\mathbf{M}_i$ has corrupted VC-dimension $d_i$ and $\mathbf{M}_i^\top$ has corrupted VC dimension $d_i'$ for $i \in \{1,2\}$. Let $s = \min\{d_1, d_2, d_1', d_2'\}$. Then, there is an algorithm for Boolean matrix multiplication which runs in $\widetilde{O}(n^{3-1/s})$ time.*

Another relevant conjecture is the OMv conjecture, which states that no algorithm can preprocess a Boolean matrix $M \in \{0,1\}^{n \times n}$ in polynomial time, and then answer $n$ queries that compute $\mathbf{M}v^{(i)}$ for an online sequence of Boolean vectors $v^{(1)}, ..., v^{(n)} \in \{0,1\}^n$ in total time $O(n^{3-\epsilon})$. Observe

that here the vector $v^{(i+1)}$ is only given after the data structure returned $\mathbf{M}v^{(i)}$, hence the use of fast matrix multiplication to compute the matrix product $\mathbf{M}\,[v^{(1)}|...|v^{(n)}]$ is ruled out.

This conjecture is used to prove conditional lower bounds for many dynamic graph problems, such as maintaining shortest paths, effective resistances, reachability, bipartite matching, triangle detection, and many others in graphs undergoing vertex insertions and deletions [Henzinger et al., 2015]. Even using fast matrix multiplication, no algorithm can beat $O(n^2)$ time on dense graphs, which is typically equivalent to re-solving them from scratch whenever the graph changes. Since Theorem 1.3 shows the OMv conjecture does not hold for structured matrices, it leads to improved dynamic algorithms.

**Implications to Dynamic Algorithms.** We list some novel dynamic algorithms obtained via this technique, whose upper bounds are only achievable due to their structured nature, otherwise violating either the BMM or OMv conjecture. We defer the formal proofs for these results to Appendix C.

An important class of linear equations arising in practice have form $\mathbf{L}x = b$, where $\mathbf{L}$ is the Laplacian of an undirected graph $G = (V, E)$ given by $\mathbf{L} := \mathbf{D} - \mathbf{A}$ where $\mathbf{D}$ is a diagonal matrix such that $\mathbf{D}_{i,i} = \deg(i)$ for $i \in [n]$ where $|V| = n$, and $\mathbf{A}$ is the adjacency matrix of $G$. There are Laplacian solvers that run in time nearly linear in $|E|$ [Spielman and Teng, 2004]. We study the problem of constructing a dynamic Laplacian solver for $\mathbf{L}_t x = b_t$ where the Laplacian changes over time through vertex updates to the graph, and each $b_t$ is an arbitrary new vector.

Previous upper bounds could only handle a $\mathrm{poly}(1/\epsilon)$ accuracy by maintaining spectral sparsifiers [Abraham et al., 2016] or vertex sparsifiers [Durfee et al., 2019]. Moreover, exact or $O(\log 1/\epsilon)$-time dependencies are ruled out by the OMv conjecture since no algorithm beats naive recomputation from scratch in $\widetilde{O}(|E|) = \widetilde{O}(n^2)$ time via nearly-linear time Laplacian solvers [Spielman and Teng, 2004]. We give the first result faster than naive recomputation in the high-accuracy $\log(1/\epsilon)$ regime.

**Theorem 1.6.** *(Dynamic Laplacian Solver). There is a dynamic algorithm that, given a dynamic graph $G = (V, E)$ with corrupted VC-dimension bounded by $d$, maintains a Laplacian system solver. The data structure supports queries that receive a vector $b \in \mathbb{R}^{|V|}$ and error parameter $\epsilon > 0$. Then, in $\widetilde{O}(n^{2-1/d} \log 1/\epsilon)$ time, the algorithm returns the (approximate) solution $x$ to $\mathbf{L}x^* = b$ where $\|x - x^*\|_{\mathbf{L}} \leq \epsilon \|x^*\|_{\mathbf{L}}$. Each vertex update to $G$ takes $\widetilde{O}(n)$ time.*

With the fast dynamic Laplacian solver, we maintain dynamic effective resistances, which is a critical subroutine in graph clustering [Alev et al., 2017], fault-tolerant computing [Ghosh et al., 2008], network analysis [Tizghadam and Leon-Garcia, 2008, Anand and Umans, 2023, Anand et al., 2024a], and biological systems [Klein et al., 2022], as it measures the connectivity between two vertices. The effective resistance $r_G(a, b)$ between vertices $a$ and $b$ in a graph $G$ is a graph-analog of leverage scores that represents the energy needed to route one unit of electric flow from $a$ to $b$. Formally, let $\mathbf{e}_i$ denote the $i$-th standard basis vector. Then, for all $a, b \in V$, $r_G(a, b) = (\mathbf{e}_a - \mathbf{e}_b)^\top \mathbf{L}^\dagger (\mathbf{e}_a - \mathbf{e}_b)$, where $\mathbf{L}$ is the Laplacian of $G$ and $\mathbf{L}^\dagger$ is the Moore-Penrose pseudoinverse of $\mathbf{L}$.

**Theorem 1.7.** *(Dynamic Effective Resistance). There is a dynamic algorithm that, given a dynamic graph $G = (V, E)$ with corrupted VC-dimension bounded by $d$, maintains effective resistances in $G$. The data structure supports queries that receive a pair of vertices $u, v \in V$ and error parameter $\epsilon > 0$. Then, in $\widetilde{O}(n^{2-1/d} \log 1/\epsilon)$ time, the algorithm returns a $(1 \pm \epsilon)$-approximation of the effective resistance. Moreover, each node update to $G$ in the dynamic data structure takes $\widetilde{O}(n)$ time.*

Our dynamic matrix-vector multiplication data structure also leads to faster dynamic triangle detection algorithms for graphs with corrupted VC-dimension $d$, which are critical components of fraud detection systems and community-detection algorithms. No subquadratic time algorithm exists for vertex updates, conditional on BMM [Abboud and Vassilevska Williams, 2014] or OMv [Henzinger et al., 2015]. We show that these lower bounds can be broken on structured graphs.

**Theorem 1.8.** *(Dynamic Triangle Detection). There is an algorithm that, given a dynamic graph $G = (V, E)$ with corrupted VC-dimension $d$, maintains whether $G$ has a triangle or not. Each vertex update takes $\widetilde{O}(n^{2-1/d})$ time and returns a Boolean indicator for $G$ containing a triangle.*

Our techniques lead to faster dynamic approximate algorithms for single-source distances, where we are tasked with finding the distances from a designated source vertex to every other vertex.

**Theorem 1.9.** *(Dynamic Approximate Single-Source Shortest Paths). There is a dynamic algorithm that maintains $(1 + \epsilon)$-approximate single-source distances on a dynamic unweighted graph*

$G = (V, E)$. *If the corrupted VC-dimension of $G$ is bounded by $d$, each node update to $G$ takes $\widetilde{O}(kn^{2-1/2d}/\epsilon)$ time, and querying the distances for any source node takes $\widetilde{O}(n^{2-1/2d}/\epsilon)$ time.*

Given a metric space $\mathcal{M}$ with $n$ points and a positive integer $k \le n$, the $k$-center problem asks one to select $k$ points, referred to as centers, such that the maximum distance of any point in the metric space to its closest center is minimized. It is known that the $k$-center is NP-hard to $(2 - \epsilon)$-approximate for any $\epsilon > 0$ [Hsu and Nemhauser, 1979]. The best existing dynamic algorithms [Cruciani et al., 2024] only work with edge updates and use fast matrix multiplication as a black-box. We present an algorithm without fast matrix multiplication that supports more powerful vertex updates.

**Theorem 1.10.** *(Dynamic Approximate $k$-center). Given an unweighted undirected graph $G = (V, E)$, there is a dynamic algorithm for $(2 + \epsilon)$-approximate $k$-center with node update time $\widetilde{O}(kn^{2-1/2d}/\epsilon)$, where $d$ is a bound on the corrupted VC-dimension of $G$.*

### 1.3 Further Related Work

**Boolean matrix multiplication.** A line of work shows that the complexity of matrix-vector multiplication over the Boolean semiring can be mildly subpolynomial. For instance, Williams [2007] showed that BMM can be computed in $O(n^3/\log n)$ time. In turn, Larsen and Williams [2017] and Abboud et al. [2024] extended this to $O(n^3/2^{\Omega(\sqrt[7]{\log n})})$.

**Structured matrices.** When the underlying matrix is Vandermonde, Toeplitz, Hankel, or Cauchy, a body of work shows that they admit fast matrix-vector multiplications through convolutional transformations [Motzkin, 1951, Olshevsky and Shokrollahi, 2000, Pan and Tsigaridas, 2014]. Orthogonal-vector (OV) matrices are structured matrices where each row and column receives a label $v \in \{0, 1\}^d$ and $\mathbf{M}_{i,j} = 1$ if and only if the corresponding two labels are orthogonal. Similar to our corrupted VC-dimension results, Alman and Vassilevska Williams [2020] show that OV matrices allow fast matrix-vector products if there are few corruptions to the matrix. Matrix products can also be accelerated using geometric data structures Lingas [2002], Floderus et al. [2018]. Another approach to handle structured inputs is through the algorithms with predictions regime, where some learning algorithm extracts the structure and can predict information about future vectors, leading to faster matrix-vector multiplication algorithms Henzinger et al. [2024], Brand et al. [2024].

**Dynamic structured graphs.** Our dynamic algorithms hold for any structured graph, even if we do not know their structure. When the structure is known, there are specialized dynamic algorithms tailored to the specific graph. These include, for instance, interval graphs Crespelle [2019], Chen et al. [2024], geometric graphs defined from Kernels [Alman et al., 2020], dynamic planar graphs [Korhonen et al., 2024, Abboud and Dahlgaard, 2016], and minor free graphs [Dorn et al., 2012].

**Matrix vector multiplication in optimization.** In convex optimization, there is a long history on developing data structures to accelerate matrix-vector products. Here the matrices are generally projection matrices that require computation of some matrix inverse. While historically, most research was on accelerating the maintenance of the matrix inverse [Karmarkar, 1984, Vaidya, 1989, Lee and Sidford, 2015, Cohen et al., 2021, Lee et al., 2019, Brand, 2020, Jiang et al., 2021, 2020a, Huang et al., 2022, Jiang et al., 2022, 2020b], this research has progressed so far that the current bottleneck for faster linear program solvers are simple matrix-vector products [Brand et al., 2020, 2021].

## 2 Preliminaries

**Notation.** Let $[n] = \{1, \ldots, n\}$. We use $O(\cdot)$ to hide constant factors, and $\widetilde{O}(\cdot)$ to hide polylogarithmic factors. Let $\|\cdot\|_1$ denote the $\ell_1$ (Manhattan distance). For a matrix $\mathbf{M}$, let $\mathbf{M}_i$ and $\mathbf{M}_{.;i}$ denote its $i$'th column/row (respectively). For a set $V$ we write $\mathbf{M} \in \mathbb{R}^{m \times V}$ for the $m \times |V|$ matrix where we can index columns by $x \in V$, i.e., $\mathbf{M}_x \in \mathbb{R}^m$. For any vector $x \in \mathbb{R}^n$, the Hamming weight of $x$ is the number of non-zero entries of $x$ also denoted by $\text{nnz}(x)$. For Boolean vectors $x, y \in \{0, 1\}^n$, the Hamming distance between $x$ and $y$ is given by $\|x - y\|_1$. Finally, for any vector $x \in \mathbb{R}^n$ and positive definite matrix $\mathbf{L} \in \mathbb{R}^{n \times n}$, define the $\mathbf{L}$-induced norm of $x$ by $\|x\|_{\mathbf{L}} = \sqrt{x^\top \mathbf{L} x}$.

**VC-Dimension.** A range space (or a set system) is a pair $\mathfrak{R} = (X, R)$ where $X$ is a set and $R$ is a set of subsets of $X$. Let $\Pi_R(A) = \{A \cap r : r \in R\}$. The VC-dimension of $\mathfrak{R}$ is $\text{VC}(\mathfrak{R}) = \max\{|S| : S \subseteq X \text{ and } |\Pi_R(S)| = 2^{|S|}\}$. The dual range space of $\mathfrak{R}$ is then given by $\mathfrak{R}^\top = (R, \{\{r | x \in r\} | x \in X\})$, and the dual VC-dimension of $\mathfrak{R}$ is the VC-dimension of $\mathfrak{R}^\top$. Any matrix $\mathbf{M} \in \{0, 1\}^{m \times n}$

corresponds to a set family $\mathcal{F}_\mathbf{M} = \{\{j : \mathbf{M}_{i,j} = 1\} : i \in [m]\}$, i.e., each row of $\mathbf{M}$ is the indicator vector of some set. Through it, we can encode $\mathbf{M}$ by the range space $\mathfrak{R}_M = ([n], \mathcal{F}_\mathbf{M})$. Then, the VC-dimension of $\mathbf{M}$ is defined as the VC-dimension of $\mathfrak{R}_\mathbf{M}$, and the dual VC-dimension is the VC-dimension of $\mathbf{M}^\top$. Here, a subset $S$ of the columns of $\mathbf{M}$ is said to be *shattered* if each of the $2^{|S|}$ many 0/1 strings appears in some row in the restriction of $\mathbf{M}$ to the columns in $S$. Then, the VC-dimension of $\mathbf{M}$ is the maximum size of a shattered subset of the columns of $\mathbf{M}$. Therefore, we see that the VC-dimension $d_{m,n}$ of a matrix $\mathbf{M} \in \{0,1\}^{m \times n}$ satisfies $1 \le d_{m,n} \le \log m$.

# 3 The Structural Complexity of Matrix-Vector Multiplication

We present a technical overview of the algorithms and main theorems. We begin this section with an overview of Björklund and Lingas [2001], Alves et al. [2024] who established a connection between matrix vector products and minimum spanning trees on $n$ points $\{0, 1\}^m$, where the edge weights are given by the Hamming distance between two points (i.e., matrix $\mathbf{M}$ defines a collection of points).

Having established this connection, we prove one of our main results in Section 3.2. We prove via techniques from computational geometry that the weight of the minimum spanning tree is bounded if the matrix $\mathbf{M}$ has corrupted VC-dimension at most $d$. This yields our main result in Theorem 1.3.

## 3.1 Matrix Vector Products via Minimum Spanning Trees

We here recap the algorithm idea by Björklund and Lingas [2001], which was later independently rediscovered by Alves et al. [2024]. We also explain how to extend the results from $\mathbf{M}$ to $\mathbf{M}^\top$.

**Differential Compression.** Given a binary matrix $\mathbf{M} \in \{0,1\}^{m \times n}$, let $\mathbf{M}_i$ denote its $i$'th column. We compress $\mathbf{M}$ by writing each column $\mathbf{M}_x$ as the sum of another column $\mathbf{M}_y$ and a *change vector* $\Delta_{x,y}$ such that $\mathbf{M}_y = \mathbf{M}_x + \Delta_{x,y}$ and $\Delta_{x,y} = \mathbf{M}_x - \mathbf{M}_y$. If $\mathbf{M}_x$ and $\mathbf{M}_y$ are similar, then $\|\Delta_{x,y}\|_1$ is likely to be smaller than the number of nonzero elements of $\mathbf{M}_x$. So, it is more efficient to indirectly represent $\mathbf{M}_x$ with respect to $\mathbf{M}_y$, instead of through $\mathbf{M}$. This requires an algorithm that finds a chain of $\Delta$'s to represent all columns of $\mathbf{M}$: for each column $\mathbf{M}_x$, identify a similar column $\mathbf{M}_y$ such that the Hamming weight $\|\Delta_{x,y}\|_1$ required to represent $\mathbf{M}_x$ is minimized subject to $\mathbf{M}_y$. Since the algorithm measures the Hamming distance for each pair of columns in $\mathbf{M}$, we model this compression by an undirected weighted graph $G$ on $n$ vertices, where vertex $i$ represents $\mathbf{M}_i$, and the weight of each edge $(x, y)$ is $\|\Delta_{x,y}\|_1$. We can then find a Minimum Spanning Tree (MST) of $G$, which by definition spans $G$ with the minimum sum of edge weights possible. Therefore, any MST of $G$ rooted at vertex $r$ defines a chain of $\Delta$'s that, starting at $r$, can represent all the columns of $\mathbf{M}$.

We formalize this intuition by defining the $\Delta$-labeled spanning tree of a matrix and its weight.

**Definition 3.1** ($\Delta$-labeled spanning tree of matrix $\mathbf{M}$). *A $\Delta$-labeled tree is a tree $T = (V, E)$ with some explicit root $x$, where each edge $e \in E$ is directed away from $x$ and labeled by a vector $\Delta_e \in \mathbb{R}^m$. We say $T$ is a $\Delta$-labeled spanning tree for matrix $\mathbf{M} \in \mathbb{R}^{m \times n}$ if $\mathbf{M}_x = \vec{0}$ for root $x$, and for all edges $(i, j) \in E$ we have $\Delta_{i,j} \coloneqq \mathbf{M}_j - \mathbf{M}_i$.*

**Definition 3.2** (Weight of a $\Delta$-labeled spanning tree). *For any $\Delta$-labeled spanning tree of $\mathbf{M}$ denoted by $T$ with directed edge-set $E$, define $\mathrm{weight}(T) = \sum_{e \in T} \mathrm{nnz}(\Delta_e)$.*

The minimum spanning tree of $\mathbf{M}$ is then the $\Delta$-labeled spanning tree with the smallest possible weight. Given *any* spanning tree of $\mathbf{M}$, we use the weight of the $\Delta$-labeled spanning tree to parameterize the runtime of the matrix-vector multiplication algorithm. Here, the lower the weight of the $\Delta$-labeled spanning tree, the faster the runtime.

**Lemma 3.3** (Björklund and Lingas [2001], Alves et al. [2024]). *Given a $\Delta$-labeled spanning tree $T$ of $\mathbf{M}^\top \in \{0,1\}^{n \times m}$ and a vector $v \in \mathbb{R}^n$, we can compute $\mathbf{M}v$ in $O(\mathrm{weight}(T) + n + m)$ time.*

We describe this procedure in Algorithm 1. As a quick recap, the algorithm computes $\mathbf{M}v$ by starting at a root node $r \in T$ of the spanning tree and performing depth-first search (DFS). The algorithm initializes an output vector and computes $(\mathbf{M}v)_r$ in $O(n)$ time. For each edge $(x, y)$ traversed by DFS on the tree, the algorithm computes $(\mathbf{M}v)_y = (\mathbf{M}v)_x + \Delta_{y,x}^\top v$. The complexity of this iterative procedure is proportional to $\mathrm{weight}(T) = \sum_{e \in E(T)} \delta_e$ where for each edge $e = (x, y)$ in the $\Delta$-labeled spanning tree, $\delta_e$ is the number of non-zeros in $\Delta_e$, i.e., the Hamming distance between row $x$ and row $y$ of $\mathbf{M}$. For completeness, we prove Lemma 3.3 in Appendix B.1.

Since the weight of the $\Delta$-labeled spanning tree of $\mathbf{M}^\top$ can be different from the weight of the $\Delta$-labeled spanning tree of $\mathbf{M}$, we provide a new algorithm with an analogous guarantee in Lemma 3.4.

**Lemma 3.4.** *Given a $\Delta$-labeled spanning tree $T$ of $\mathbf{M} \in \{0,1\}^{m \times n}$ and a vector $v \in \mathbb{R}^n$, there is an algorithm that can compute $\mathbf{M}v$ in $O(\text{weight}(T) + n + m)$ time.*

To utilize the $\Delta$-labeled spanning tree of $\mathbf{M}$, note that $\mathbf{M}v = \sum_{i=1}^n \mathbf{M}_i v_i$. In the special case where $\mathbf{M}$ has only two columns, $\mathbf{M}v = \mathbf{M}_1 v_1 + \mathbf{M}_2 v_2 = \mathbf{M}_1(v_1 + v_2) + \Delta_{1,2} v_2$. For more columns, this can be extended to require only a product with $\mathbf{M}_1$ and products with each $\Delta_e$ in $T$. Thus the complexity is bounded by $O(m + n + \text{weight}(T))$. We prove Lemma 3.4 in Appendix B.1.

To minimize the time complexity, we compute a MST. In Alves et al. [2024] this was done naïvely, whereas in Björklund and Lingas [2001] this is done via Theorem 4.2 of Indyk and Motwani [1998].

**Lemma 3.5** (Theorem 4.2 of Indyk and Motwani [1998]). *For $\epsilon > 0$, we can compute a $(1 + \epsilon)$-approximate MST with respect to the Hamming distance of $n$ points in $\{0,1\}^m$ in time $\widetilde{O}(mn^{1 + \frac{1}{1+\epsilon}})$.*

**Theorem 3.6.** *Given a Boolean $m \times n$ matrix $\mathbf{M}$, we can construct in $\widetilde{O}(nm)$ time a $\Delta$-labeled spanning tree whose weight is at most a $O(\log n)$-factor larger than the MST of $\mathbf{M}$.*

*Proof.* Run Lemma 3.5 on the columns of $\mathbf{M}$ for $\epsilon = \log n$ to construct an $O(\log n)$-approximate MST in $\widetilde{O}(mn)$ time. Next, iterate over the edges of the MST and label each tree edge $(x, y)$ by $\Delta_{x,y} = \mathbf{M}_y - \mathbf{M}_x$. This takes $\widetilde{O}(mn)$ time as there are only $n - 1$ tree edges and $\Delta_{x,y} \in \mathbb{R}^m$. $\square$

### 3.2 Existence of Low Weight $\Delta$-labeled Minimum Spanning Trees

The Hamming distance between any two columns in $\mathbf{M}$ is at most $m$. Since there are $O(n)$ edges in the $\Delta$-labeled MST, this gives a coarse bound of $O(mn)$ which is achieved for Hadamard matrices. One of our main technical contributions is to prove that $\Delta$-labeled MST's with low weight exist for structure matrices. We bound the MST's weight parameterized by the corrupted VC-dimension of $\mathbf{M}$.

**Theorem 3.7.** *If a matrix $\mathbf{M}^\top \in \{0,1\}^{n \times m}$ has corrupted VC-dimension $d$, then the weight the minimum spanning tree of $\mathbf{M}$ is at most $O(mn^{1-1/d} \log^2 n)$.*

We show that for matrices $\mathbf{M}$ whose transpose $\mathbf{M}^\top$ has bounded corrupted VC-dimension, the MST has small weight. For this, we work directly with the definition of VC-dimension and use set terminology, which allows us to use relevant work from Welzl [1988]. This previous work studied bounded VC-dimensions in the context of computational geometry. We here use their results to bound the complexity of matrix vector products.

**Definition 3.8** (Set crossings). *Let $A$ and $r$ be sets. $r$ crosses $A$ if and only if neither $A \subseteq r$ nor $A \cap r = \emptyset$ holds. That is, there are $x, y \in A$ with $x \in r$ and $y \notin r$.*

**Definition 3.9** (Spanning Tree). *Let $A \subset X$ be a finite set of elements of a range space $(X, R)$. A spanning tree on $A$ is an undirected (unrooted) tree with node set $A$.*

**Definition 3.10** (Weight of spanning tree $T$). *A crossing of a range $r \in R$ in $T$ is an edge in $T$ that is crossed by $r$ (i.e., an edge that has one endpoint in $r$ and the other endpoint in $X \setminus r$). The crossing number $\kappa_r(T)$ of a range $r \in R$ is the number of crossings of $r$ in $T$. The weight of tree $T$ is defined as $w(T) = \sum_{r \in R} \kappa_r(T)$.*

This spanning tree corresponds to the $\Delta$-labeled spanning tree for matrix $\mathbf{M}$, where the range space $(X, R)$ induced by $\mathbf{M} \in \{0,1\}^{m \times n}$ has ground set $X = [n]$ (i.e., column indices) with the rows of $\mathbf{M}$ being the indicator vectors of $m$ subsets of $X$. Thus, a spanning tree $T_X$ for $(X, R)$ and a spanning tree $T_M$ for $\mathbf{M}$ both consist of edges connecting elements in $[n]$, and the number of crossings for an edge $(x, y) \in T_X$ equals the number of indices $i$ where $(\mathbf{M}_x)_i$ and $(\mathbf{M}_y)_i$ differ. Hence, $w(T_X)$ is the weight of the corresponding $\Delta$-labeled spanning tree of $\mathbf{M}$.

We show that when the dual of $(X, R)$ has corrupted VC-dimension $d$, then $(X, R)$ has spanning trees with small weight. The literature on finding spanning trees on $X$ with few crossings stems from works in computational geometry from the 1900s [Welzl, 1988, B. Chazelle, 1989, Matoušek, 1991, Har-Peled, 2009] which used the trees to preprocess a set of points in $\mathbb{R}^d$ and answer queries where for any convex polytope, the query returns a point in the polytope. We transfer their techniques to our problem. The following lemma allows us to prove the existence of a $\Delta$-labeled spanning tree for matrices whose transpose has bounded VC-dimension. We later extend this to corrupted VC-dimension.

**Lemma 3.11** (Lemma 4.1 of Welzl [1988]). *Let $(X, R)$ be a range space with dual VC-dimension at most $d$. For every $A \subseteq X, |A| = n \geq 1$ and every multiset $Q$ of ranges in $R$ with $|Q| = m$, there exists $x \neq y \in A$ such that the number of ranges in $Q$ crossing $(x, y)$ is at most $O(mn^{-1/d} \log^2 n)$.*

**Remark 3.12.** *The original proof of Lemma 4.1 of Welzl [1988] gives a bound of $O(mn^{-1/d} \log n)$ but hides some terms that depend on the VC-dimension: the original statement of Lemma 4.1 says that the constant c, hidden in the $O(\cdot)$ notation, depends on $(X, R)$, but this is vacuous in our setting as $(X, R)$ is the entire problem instance. We show that $c = O(d) \leq O(\log n)$ suffices, and provide a proof in B.2 for completeness.*

**Lemma 3.13.** *Let $(X, R)$ be a range-space with $|X| = n, |R| = m$ with dual VC-dimension at most $d$. Then there exists a spanning tree $T$ of weight $w(T) = O(mn^{1-1/d} \log^2 n)$.*

*Proof.* Let $A_0 = X$. Consider the following process initialized at $i = 0$. By Lemma 3.11, there exists distinct $x_i, y_i \in A_i$ where the number of ranges in $R$ crossing $(x_i, y_i)$ is $O(m|A_i|^{-1/d} \log^2 n)$. Iteratively letting $A_{i+1} \leftarrow A_i \setminus x_i$ constructs a spanning tree $T$ with weight at most

$$\sum_{i=1}^{n} O(m(n-i)^{-\frac{1}{d}} \log^2(n-i)) \leq O\left(m \log^2 n \int_1^n x^{-\frac{1}{d}} \mathrm{d}x\right) = O(mn^{1-\frac{1}{d}} \log^2 n). \qquad \square$$

We extend Lemma 3.13 to corrupted VC-dimensions by converting the tree to a single line of vertices, so that any single corruption minimally affects the number of crossings.

For instance, suppose $X = [n]$, $R$ is a multiset of $m$ empty ranges (corresponding to an $m \times n$ zero matrix $\mathbf{M}$), and the spanning tree is a star with $1 \in X$ at the center. If the corruption adds 1 to each empty set in $R$ (equivalently, turning the first column of $\mathbf{M}$ to an all-1-column), this fits into the definition of the dual having corrupted VC-dimension. This is because each set of $(X, R)$ is corrupted by only one element, and thus in the dual set system each element is corrupted by adding it to one set.

After corruption, the number of crossings of each tree edge is $O(m)$ as 1 is in each of the $m$ sets, but none of the other elements in $X$ are in any set. As the tree has $n-1$ edges, each connecting 1 to another element of $X$, the tree has $O(mn)$ crossings in total. If, however, the tree was a single line, the corruption would affect at most 2 edges, and the total number of crossings would only increase by $O(m)$ rather than $O(mn)$. We convert the tree to a single line in Lemma 3.14, proven in Appendix B.2.

**Lemma 3.14.** *For a range space $(X, R)$, $|X| = n$ with spanning tree $T$, there exists a permutation $\pi \in S_n$ such that the weight of the corresponding spanning tree $T'$ (which is just the line $\pi(1), \pi(2), ..., \pi(n)$) is at most twice that of $T$.*

*Proof of Theorem 3.7.* Suppose $\mathbf{M}^\top = \mathbf{L}^\top + \mathbf{S}^\top \in \{0,1\}^{n \times m}$ has corrupted VC-dimension $\leq d$, where $\mathbf{L}^\top \in \{0,1\}^{n \times m}$ has VC-dimension $d$ and $\mathbf{S}^\top \in \{-1,0,1\}^{n \times m}$ has at most $O(n^{1-1/d})$ non-zero entries per column. $\mathbf{L}$ and $\mathbf{S}$ exist by definition of corrupted VC-dimension.

Let $(X, R)$ be the set system induced by $\mathbf{L}$ (i.e., each row of $\mathbf{L}$ is the indicator vector of a set in $R$ with $|R| = m$), by $\mathbf{L}^\top$ having VC-dimension $\leq d$, $(X, R)$ has dual VC-dimension $\leq d$. By Lemma 3.13, $(X, R)$ has a spanning tree $T$ with $w(T) \leq O(mn^{1-1/d} \log^2 n)$. Then, by Lemma 3.14, there exists a spanning tree $T'$ which is a line with weight at most $O(mn^{1-1/d} \log^2 n)$. Let $(X, R_c)$ be the range space of $\mathbf{M}$. $T'$ has the same weight as $(X, R_C)$ as on $(X, R)$, excluding crossings due to the corruption. For range $r \in R_c$, we count the number of new crossings it incurs due to the corruption: an edge $(x, y)$ only has a new crossing for a range $r \in R_c$ if $x$ or $y$ was corrupted in $r$. Since each $x \in X$ has at most 2 incident edges in $T'$ (since it is a line) each corruption to $r$ causes at most 2 new crossings. Since the total number of corruptions is bounded by $O(mn^{1-1/d})$, we have $w(T') \leq O(mn^{1-1/d} \log^2 n)$ and $T'$ is a $\Delta$-labeled spanning tree for $\mathbf{M}$, as $X = [n]$ are the column indices and the number of crossings for an edge $(x, y)$ is the number of entries where $\mathbf{M}_x$ and $\mathbf{M}_y$ differ. Since the MST has lower weight, its weight is at most $O(mn^{1-1/d} \log^2 n)$. $\qquad \square$

We can now put everything together to prove Theorem 1.3.

**Theorem 1.3.** *(Static Online Matrix-Vector Multiplication). If a matrix $\mathbf{M} \in \{0,1\}^{m \times n}$ has corrupted VC-dimension $d$, then after an $\widetilde{O}(mn)$-time preprocessing, there is a data structure $\mathcal{D}$ that can compute $\mathbf{M}v$ for any $v \in \mathbb{R}^n$ in $\widetilde{O}(nm^{1-1/d} + m)$ time, with high probability.*

*Proof.* Preprocess $\mathbf{M}$ by constructing a $\Delta$-labeled spanning tree $T$ for $\mathbf{M}^{\top}$ via Theorem 3.6 in $\widetilde{O}(mn)$ time. Given $v \in \mathbb{R}^n$, pass $v$ and $T$ to the algorithm of Lemma 3.3. It returns $\mathbf{M}v$ in time $\widetilde{O}(m+n+\text{weight}(T))$. By Theorem 3.7, the MST of $\mathbf{M}^{\top}$ has weight at most $\widetilde{O}(nm^{1-1/d})$ since $\mathbf{M}$ has corrupted VC-dimension at most $d$. Since $T$ is a $O(\log n)$-approximation of the MST, answering a query takes $\widetilde{O}(nm^{1-1/d} + m)$ time. $\qquad\square$

### 3.3 Dynamic Matrices

At last, we outline how to handle matrices $\mathbf{M}$ that receive updates over time, i.e., Theorem 1.4.

**Theorem 1.4.** *(Dynamic Online Matrix-Vector Multiplication). Given a matrix $\mathbf{M} \in \{0,1\}^{m \times n}$, there is a data structure $\mathcal{D}$ with $\widetilde{O}(mn)$ preprocessing time that supports row and column updates (insertions/deletions) to $\mathbf{M}$ in $\widetilde{O}(n)$ and $\widetilde{O}(m)$ time, respectively. Upon querying $\mathcal{D}$ with a vector $v \in \mathbb{R}^n$, it outputs $\mathbf{M}v$ in $\widetilde{O}(nm^{1-1/d^*} + m)$ time, with high probability, where $d^*$ is the largest corrupted VC-dimension of $\mathbf{M}$ throughout the history of its updates.*

We sketch how to handle column insertions: Let $\mathbf{M}$ be the current matrix. We split it into smaller matrices $\mathbf{M}^{(0)}, \mathbf{M}^{(1)}, ..., \mathbf{M}^{(\log n)}$, where each $\mathbf{M}^{(i)}$ contains at most $2^i$ columns of $\mathbf{M}$. When a new column is inserted to $\mathbf{M}$, we insert that column to $\mathbf{M}^{(0)}$. When any $\mathbf{M}^{(i)}$ contains more than $2^i$ columns, then all its columns are inserted to $\mathbf{M}^{(i+1)}$ and $\mathbf{M}^{(i)}$ is reset to an empty matrix. Thus, any $\mathbf{M}^{(i)}$ receives new columns at most once every $2^i$ iterations to $\mathbf{M}$. When an $\mathbf{M}^{(i)}$ receives new columns, we initialize a new matrix-vector multiplication data structure (Theorem 1.3) on it. This takes $\widetilde{O}(m2^i)$ time, but amortizes to $\widetilde{O}(m)$ time per insertion to $\mathbf{M}$ since it only happens at most once every $2^i$ insertions. This yields an amortized time of $\sum_{i=0}^{\log n} \widetilde{O}(m) = \widetilde{O}(m)$ time to maintain all $\mathbf{M}^{(i)}$ for $i = 0, ..., \log n$. To answer a matrix-vector multiplication query $\mathbf{M}v$, we split $v$ into vectors $v^{(0)}, ..., v^{(\log n)}$ such that $\mathbf{M}v = \sum_{i=0}^{\log n} \mathbf{M}^{(i)} v^{(i)}$. For the time complexity, note that each $\mathbf{M}^{(i)}$ is a submatrix of $\mathbf{M}$. Thus, the the time complexity needed to multiply a vector by $\mathbf{M}^{(i)}$ is bounded by $\widetilde{O}(nm^{1-1/d} + m)$.

We ignore column deletions to $\mathbf{M}^{(i)}$ until $\mathbf{M}^{(i)}$ is reinitialized: simply set $v_j = 0$ for the columns that should have been deleted. We can handle row updates to $\mathbf{M}$ in a similar way, by splitting the matrix again into $\log m$ matrices, each containing at most $2^j$ rows. The full proof is given in Appendix B.3.

## 4 Conclusion

We study the structural complexity of matrix-vector multiplication. We propose a theoretical framework to study heuristic algorithms that have achieved tremendous practical success. Building on the VC-dimension literature, we propose the notion of a corrupted VC-dimension $d$ and provide structural characterizations. We show that if a matrix $\mathbf{M} \in \{0,1\}^{m \times n}$ has corrupted VC-dimension $d$, then its matrix-vector product $\mathbf{M}v$ can be computed in $\widetilde{O}(nm^{1-1/d} + m)$ time, providing polynomial speed-ups over existing methods. Moreover, our algorithm maintains updates to $\mathbf{M}$ in $\widetilde{O}(m)$ and $\widetilde{O}(n)$ time, leading to the first $\widetilde{O}(n^{2-\epsilon})$-time algorithms for maintaining high-accuracy estimates for Laplacian solvers, effective resistance, and triangle detection in dynamic graphs.

**Limitations.** While we show subquadratic runtimes in non-Boolean matrices with bounded Pollard pseudodimension, we believe that the linear dependence of the number of distinct values $T$ in the runtime is suboptimal and should instead be logarithmic. Finally, without matching lower bounds, we cannot verify whether the $\widetilde{O}(n^{2-1/d})$ scaling with respect to the VC-dimension is optimal.

**Societal Impacts.** This work is foundational in nature. As such, while it enables more efficient computation in iterative algorithms, it is not tied to any specific applications or deployments.

## Acknowledgements

This work was supported by NSF Grants DMS 2202961 and CCF 2338816. We also express our thanks to Ce Jin, Andrzej Lingas, Albert Weng, Jesper Jansson, Shunhua Jiang, Xiaobai Sun, Elia Gorokhovsky, and Jingtong Sun for insightful discussions.

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

# Contents

# A   Preliminaries for Supplementary Material

**Notation.** Let $[n] = \{1, \ldots, n\}$. We use $O(\cdot)$ to hide constant factors, and $\widetilde{O}(\cdot)$ to hide polylogarithmic factors. Let $\| \cdot \|_1$ denote the $\ell_1$ (Manhattan distance). For any matrix $\mathbf{M}$, let $\mathbf{M}_i$ and $\mathbf{M}_{:,i}$ denote its $i$'th column and $i$'th row (respectively). For a set $V$ we write $\mathbf{M} \in \mathbb{R}^{m \times V}$ for the $m \times |V|$ matrix where we can index columns by $x \in V$, i.e., $\mathbf{M}_x \in \mathbb{R}^m$. For any vector $x \in \mathbb{R}^n$, the Hamming weight of $x$ is the number of non-zero entries of $x$ also denoted by $\text{nnz}(x)$. For Boolean vectors $x, y \in \{0, 1\}^n$, the Hamming distance between $x$ and $y$ is given by $\|x - y\|_1$. Finally, for any vector $x \in \mathbb{R}^n$ and positive definite matrix $\mathbf{L} \in \mathbb{R}^{n \times n}$, define the $\mathbf{L}$-induced norm of $x$ by $\|x\|_\mathbf{L} = \sqrt{x^\top \mathbf{L} x}$.

**VC-Dimension of Range Spaces.** A range space (also called a set system) is a pair $\mathfrak{R} = (X, R)$ where $X$ is a set and $R$ is a set of subsets of $X$. Let $\Pi_R(A) = \{A \cap r : r \in R\}$. The VC-dimension of $\mathfrak{R}$ is given by $\text{VC}(\mathfrak{R}) = \max\{|S| : S \subseteq X \text{ and } |\Pi_R(S)| = 2^{|S|}\}$. The primal shatter function $\pi$ of $\mathfrak{R}$ is given by $\pi(m) = \max\{|\Pi_R(A)| : A \subseteq X, |A| = m\}$ for $m \geq 0$. Let $B$ be a set of ranges in $R$. Then, let $\Pi_X^*(B) = \{C : C \subseteq X \text{ and } C \text{ is not crossed by any range in } Q\}$ and let $\Pi_X^*(B)$ be maximal. Then, for $m \geq 0$, the dual shatter function $\pi^*$ of the range space is given by $\pi^*(m) = \max\{|\Pi_X^*(B)| : B \subseteq R, |B| = m\}$.

**VC-Dimension of Matrices.** Any matrix $\mathbf{M} \in \{0, 1\}^{m \times n}$ corresponds to a set family given by $\mathcal{F}_\mathbf{M} = \{\{j : \mathbf{M}_{i,j} = 1\} : i \in [m]\}$, i.e., consider each row of $\mathbf{M}$ as the indicator vector of some set. Through it, we can encode $\mathbf{M}$ by the range space $\mathfrak{R}_M = ([n], \mathcal{F}_\mathbf{M})$. Then, the VC-dimension of $\mathbf{M}$ is defined as the VC-dimension of $\mathfrak{R}_\mathbf{M}$. Here, a subset $S$ of the columns of $\mathbf{M}$ is said to be *shattered* if each of the $2^{|S|}$ many 0/1 strings appears in some row in the restriction of $\mathbf{M}$ to the columns in $S$. Then, the VC-dimension of $\mathbf{M}$ is the maximum size of a shattered subset of the columns of $\mathbf{M}$. Moreover, the VC-dimension $d_{m,n}$ of a matrix $\mathbf{M} \in \{0, 1\}^{m \times n}$ satisfies $1 \leq d_{m,n} \leq \log m$.

**Pollard Pseudodimension of Matrices.** For a family of functions $\mathcal{F}$ mapping $X$ to $Y$, the Pollard pseudodimension of $\mathcal{F}$, denoted by $\text{Pdim}(\mathcal{F})$, is the VC dimension of the set system $(X \times Y, \{\{(x, y) \mid f(x) \geq y\} \mid f \in \mathcal{F}\})$.

For a real matrix $\mathbf{M}$, define the pseudo-dimension of $\mathbf{M}$, denoted by $\text{Pdim}(\mathbf{M})$ by thinking of the rows of $\mathbf{M}$ as functions and taking the pseudo-dimension of the resulting class of functions. Specifically, if $\mathbf{M}$ is an $m \times n$ matrix, for each $i \in \{1, \ldots, m\}$, define $f_{\mathbf{M},i} : \{1, \ldots, n\} \to \mathbb{R}$ by $f_{\mathbf{M},i}(j) = \mathbf{M}_{i,j}$, and let $\text{Pdim}(\mathbf{M}) = \text{Pdim}(\{f_{\mathbf{M},i} : i \in \{1, \ldots, m\}\})$.

# B   Online Matrix Vector Data Structure

In this section we prove all the intermediate lemmas required for our first two main results on subquadratic online matrix-vector multiplication. The proof that combined the technical lemmas was already given in Section 3.

**Theorem 1.3.** *(Static Online Matrix-Vector Multiplication). If a matrix $\mathbf{M} \in \{0, 1\}^{m \times n}$ has corrupted VC-dimension $d$, then after an $\widetilde{O}(mn)$-time preprocessing, there is a data structure $\mathcal{D}$ that can compute $\mathbf{M}v$ for any $v \in \mathbb{R}^n$ in $\widetilde{O}(nm^{1-1/d} + m)$ time, with high probability.*

**Theorem 1.4.** *(Dynamic Online Matrix-Vector Multiplication). Given a matrix $\mathbf{M} \in \{0, 1\}^{m \times n}$, there is a data structure $\mathcal{D}$ with $\widetilde{O}(mn)$ preprocessing time that supports row and column updates (insertions/deletions) to $\mathbf{M}$ in $\widetilde{O}(n)$ and $\widetilde{O}(m)$ time, respectively. Upon querying $\mathcal{D}$ with a vector $v \in \mathbb{R}^n$, it outputs $\mathbf{M}v$ in $\widetilde{O}(nm^{1-1/d^*} + m)$ time, with high probability, where $d^*$ is the largest corrupted VC-dimension of $\mathbf{M}$ throughout the history of its updates.*

## B.1   Static Online Matrix Vector Multiplication Data Structure

**Definition 3.1** ($\Delta$-labeled spanning tree of matrix $\mathbf{M}$). *A $\Delta$-labeled tree is a tree $T = (V, E)$ with some explicit root $x$, where each edge $e \in E$ is directed away from $x$ and labeled by a vector $\Delta_e \in \mathbb{R}^m$. We say $T$ is a $\Delta$-labeled spanning tree for matrix $\mathbf{M} \in \mathbb{R}^{m \times n}$ if $\mathbf{M}_x = \vec{0}$ for root $x$, and for all edges $(i, j) \in E$ we have $\Delta_{i,j} := \mathbf{M}_j - \mathbf{M}_i$.*

**Definition 3.2** (Weight of a $\Delta$-labeled spanning tree). *For any $\Delta$-labeled spanning tree of $\mathbf{M}$ denoted by $T$ with directed edge-set $E$, define $\text{weight}(T) = \sum_{e \in T} \text{nnz}(\Delta_e)$.*

---

**Algorithm 1** Online Matrix-vector multiplication (given a $\Delta$-labeled tree of $\mathbf{M}^\top$)

---

**Require:** Input matrix $\mathbf{M} \in \{0,1\}^{m \times n}$, and a $\Delta$-labeled tree of $\mathbf{M}^\top$ denoted by $T$ and rooted at $r$.
**Require:** Input vector $v \in \mathbb{R}^n$.
  **Initialize:** $\Phi = 0^m$
  $\Phi_r = \mathbf{M}_r^\top v$
  Run DFS on $T$ from root $r$. Whenever DFS visits a vertex $x \in V$ via some edge $(y, x)$, then
$$\text{set } \Phi_x = \Phi_y + \Delta_{x,y} v$$

  Return $\Phi$

---

**Lemma 3.3** (Björklund and Lingas [2001], Alves et al. [2024]). *Given a $\Delta$-labeled spanning tree $T$ of $\mathbf{M}^\top \in \{0,1\}^{n \times m}$ and a vector $v \in \mathbb{R}^n$, we can compute $\mathbf{M}v$ in $O(\mathrm{weight}(T) + n + m)$ time.*

*Proof.* Consider Algorithm 1 which takes as input $\mathbf{M} \in \{0,1\}^{m \times n}$, a $\Delta$-labeled spanning tree of $\mathbf{M}^\top$ (denoted by $T$), and a vector $v \in \mathbb{R}^n$.

We inductively prove that $\Phi = \mathbf{M}v$ over the number of visited vertices. This is true when only one vertex is visited, as that is the root $r$ and by definition $\Phi_r = (\mathbf{M}v)_r$. For the inductive step, assume DFS visits the $(k+1)$st vertex $x$. Let $y$ be the parent of the vertex, then the algorithm assigns

$$\begin{aligned}
\Phi_x &= \Phi_y + \Delta_{y,x}^\top v \\
&= (\mathbf{M}v)_y + (\mathbf{M}_x - \mathbf{M}_y)v \\
&= (\mathbf{M}v)_u.
\end{aligned}$$

Here $\Phi_y$ is already computed since it was among the first $k$ visited vertices.

**Runtime analysis**. Let $\delta_{i,j}$ be the number of non-zero entries in $\Delta_{i,j}$ for all $(i,j) \in E(T)$. Initializing the vector $\Phi$ and populating its first element $\Phi_r$ takes $O(n+m)$ time. The cost of populating the remaining entries is proportional to the number of non-zero entries in each $\Delta_{x,y}$, since we can store each $\Delta$ via a sparser representation of only its non-zero entries and use this to compute $\Delta_{y,x}^\top v$ in time $O(\delta_{y,x})$. Thus, the total time complexity is $O(\sum_{(x,y) \in E(T)} \delta_{x,y}) = O(\mathrm{weight}(T))$. Hence, the runtime of Algorithm 1 is $O(\mathrm{weight}(T) + n + m)$, proving the lemma. $\qquad\square$

When the spanning tree is constructed over the columns of $\mathbf{M}$, we run a different algorithm. The following matrix $\mathbf{N}$ is defined based on the spanning tree. We will prove (i) that multiplying vectors with $\mathbf{N}$ can be done efficiently (Lemma B.2), and (ii) that $\mathbf{N} = \mathbf{M}$ (Lemma B.3). Combining the two then yields Lemma 3.4.

**Definition B.1** (Path-$\Delta$ matrix $\mathbf{N}^{(T)}$). *Given a $\Delta$-labeled subtree $T = (V, E)$ with root $r$, define the path-$\Delta$ matrix $\mathbf{N}^{(T)} \in \mathbb{R}^{m \times V}$ such that for $u \in V$,*

$$\mathbf{N}_u^{(T)} := \sum_{e \in P(r \to u)} \Delta_e.$$

*Here, $P(r \to u)$ denotes the edges of the unique (since $T$ is acyclic) directed path from $r$ to $u$.*

**Lemma B.2.** *For $\Delta$-labeled tree $T = (V, E)$, and a vector $v \in \mathbb{R}^V$, let $\vec{\sigma} \in \mathbb{R}^E$ with $\vec{\sigma}_{(y,c)} = \sum_{z \in V(T_c)} v_z$, where $T_c$ is the subtree rooted at $c \in V$. Let $\vec{\Delta} \in \mathbb{R}^{m \times E}$ be the matrix with columns being the labels of tree $T$. Then $\vec{\Delta}\vec{\sigma} = \mathbf{N}^{(T)}v$.*

*Proof.* By the definition of $\mathbf{N}^{(T)}$, we have

$$\begin{aligned}
\mathbf{N}^{(T)}v &= \sum_{y \in V} \mathbf{N}_y^{(T)} \cdot v_y \\
&= \sum_{y \in V} \sum_{e \in P(r \to y)} \Delta_e \cdot v_y
\end{aligned}$$

Now consider how often any one edge $e$ shows up in this sum: an edge $e = (x, z)$ is in $P(r \to y)$ if and only if $z$ is an ancestor of $y$ (or $y = z$) (or, in other words, if $y$ is a descendant of $z$ (or $y = z$)).

This condition can be written as $y \in T_z$. Thus we write the sum as

$$\sum_{y \in V} \sum_{e \in P(r \to y)} \Delta_e \cdot v_y = \sum_{(x,z) \in E} \sum_{y \in T_z} \Delta_{x,z} \cdot v_y$$

$$= \sum_{(x,z) \in E} \Delta_{x,z} \sum_{y \in T_z} v_y$$

$$= \sum_{(x,y) \in E} \Delta_{x,y} \vec{\sigma}_y$$

$$= \vec{\Delta}\vec{\sigma},$$

which proves the claim. $\qquad\square$

**Lemma B.3.** *Let $T$ be a $\Delta$-labeled spanning tree of $\mathbf{M}$ where the root $r$ of $T$ has $\mathbf{M}_r = \vec{0}$, then $\mathbf{N}^{(T)} = \mathbf{M}$.*

*Proof.* We prove this by induction over the width of $\mathbf{M}$.

**Base Case.** For width 1, $T$ is just the root $r$, so $\mathbf{N}^{(T)} = \vec{0} = \mathbf{M}$.

**Inductive Step.** Suppose the statement holds for matrices of width $< n$ and we now have a matrix $\mathbf{M}$ of width $n$. Let $L \subset V$ be the leaves of $T$. Let $T'$ be the tree $T$ with these leafs removed, and let $\mathbf{M}' \in \{0,1\}^{m \times (V \setminus L)}$ be matrix $\mathbf{M}$ with all columns corresponding to $L$ removed. Then for any $y \in V \setminus L$

$$\mathbf{M}_y = \mathbf{M}'_y \overset{(1)}{=} \mathbf{N}_y^{(T')} \overset{(2)}{=} \mathbf{N}_y^{(T)}$$

where equality (1) uses the induction hypothesis and equality (2) uses the fact that any path in the leaf-removed tree $T'$ also exists in $T$.

For any $y \in L$ let $z$ be its ancestor in $T$. Therefore, $z \notin L$. Then, we have

$$\mathbf{N}_y^{(T)} = \mathbf{N}_z^{(T)} + \Delta_{z,y} \tag{1}$$

$$= \mathbf{N}_z^{(T')} + \Delta_{z,y} \tag{2}$$

$$= \mathbf{M}'_z + \Delta_{z,y} \tag{3}$$

$$= \mathbf{M}_z + (\mathbf{M}_y - \mathbf{M}_z) = \mathbf{M}_y,$$

where equation 1 uses the fact that the path from root $r$ to $y$ is a path from root $r$ to $z$ with one extra edge $(z,y)$. Equation 2 uses the fact the path from $r$ to $z$ in $T$ also exists in $T'$, and finally equation 3 follows by the induction hypothesis. Together, they prove the lemma. $\qquad\square$

**Lemma 3.4.** *Given a $\Delta$-labeled spanning tree $T$ of $\mathbf{M} \in \{0,1\}^{m \times n}$ and a vector $v \in \mathbb{R}^n$, there is an algorithm that can compute $\mathbf{M}v$ in $O(\text{weight}(T) + n + m)$ time.*

*Proof.* **Algorithm:**

We compute $\mathbf{N}^{(T)}v$ via Lemma B.2 which requires us to compute the vector $\vec{\sigma} \in \mathbb{R}^E$ with $\vec{\sigma}_{(y,c)} = \sum_{z \in V(T_c)} v_z$ for all $(y,c) \in E$ where $T_c$ is the subtree of $T$ rooted at $c$. This can be done in $O(n)$ time via a simple "dynamic programming on trees" algorithm: For any $c \in V$, let $C$ be its children. Then,

$$\vec{\sigma}_{(y,c)} = \sum_{z \in V(T_c)} v_z$$

$$= v_c + \sum_{x \in C} \sum_{z \in V(T_x)} v_z$$

$$= v_c + \sum_{x \in C} \vec{\sigma}_{(c,x)}.$$

So we can compute $\vec{\sigma}$ in $O(n)$ by starting at the leaf-edges, then propagating the sums up towards the root. At last, we multiply $\vec{\Delta}\vec{\sigma}$ in $O(\text{weight}(T))$ time as that is the number of non-zeros in $\vec{\Delta}$ (which is the matrix consisting of the tree labels as column-vectors).

**Correctness.** Without loss of generality, assume that for root $r$ of tree $T$ we have $\mathbf{M}_r = \vec{0}$. Otherwise, append a $\vec{0}$-column to $\mathbf{M}$, and add edge $(n+1, r)$ to $T$ with label $\Delta_{n+1,r} = \mathbf{M}_r - \mathbf{M}_{n+1} = \mathbf{M}_r$, and make $n+1$ the new root of $T$. This increases the tree weight by at most $m$, resulting in an additive $O(m)$ in the time complexity.

Append another entry to $v$ so that the dimensions match and $\mathbf{M}v$ is well-defined. By Lemma B.3, we thus have $\mathbf{N}^{(T)}v = \mathbf{M}v$. $\qquad\square$

## B.2 Existence of Low Weight $\Delta$-labeled Minimum Spanning Trees

We first provide a proof of Lemma 3.11 (Lemma 4.1 of Welzl [1988]), restated below for completeness.

**Lemma 3.11** (Lemma 4.1 of Welzl [1988]). *Let $(X, R)$ be a range space with dual VC-dimension at most $d$. For every $A \subseteq X$, $|A| = n \geq 1$ and every multiset $Q$ of ranges in $R$ with $|Q| = m$, there exists $x \neq y \in A$ such that the number of ranges in $Q$ crossing $(x, y)$ is at most $O(mn^{-1/d} \log^2 n)$.*

For this, we introduce the notion of $\epsilon$-nets and a crucial proposition of Welzl [1988] and Haussler and Welzl [1986].

**Definition B.4** ($\epsilon$-net). *Let $(X, R)$ be a range space and $0 \leq \epsilon \leq 1$. Let $A$ be a finite multiset of elements in $X$. Then, $N \subseteq A$ is an $\epsilon$-net of $A$ for $R$ if $\frac{|A \cap r|}{|A|} > \epsilon$ implies $r \cap N \neq \emptyset$ for all $r \in R$.*

**Proposition B.5** ([Welzl, 1988, Proposition 2.4]). *Let $(X, R)$ be a range space of VC-dimension $d \geq 1$ and let $0 < \epsilon \leq 1$ be a real number. For every finite multiset $A$ of elements in $X$, there is an $\epsilon$-net $N$ of $A$ for $R$ such that $|N| \leq \lceil \frac{8d}{\epsilon} \log \frac{8d}{\epsilon} \rceil$.*

**Lemma B.6** (Lemma 2.2 of Welzl [1988]). *If $(X, R)$ is a range space of $VC$-dimension $d$, then $(X, \widetilde{R})$, where $\widetilde{R} = \{(r' \cup r'') - (r' \cap r'') | r', r'' \in R\}$ has VC-dimension at most $O(d \log d)$.*

We are now ready to prove Lemma 3.11, which we adapt from Welzl [1988].

*Proof of Lemma 3.11.* By the pigeonhole principle, it is sufficient to show the existence of a subset $N$ of $Q$ such that (i) $\pi^*(|N|) < n$ and (ii) if a pair of elements $\{x, y\}$ is crossed by more than $dmn^{-1/d} \log n$ ranges in $Q$, then $\{x, y\}$ is crossed by a range in $N$.

For $x \in X$, let $R_x = \{r \in R : x \in r\}$ and for $x, y \in X$ with $x \neq y$, let $R_{xy}$ be the set of ranges in $R$ that cross $\{x, y\}$. Since $(R, \{R_x | d \in X\})$ is the dual of $(X, R)$ it has VC-dimension at most $d$ by assumption. So by Lemma B.6, $(R, \{R_{xy} | x, y \in X, x \neq y\})$ has VC-dimension $O(d \log d)$.

Therefore, by Proposition B.5, for every $\epsilon > 0$ and every multiset $Q$ of ranges in $R$, there is an $\epsilon$-net of $Q$ for $\{R_{xy} | x, y \in X, x \neq y\}$ with $|N| \leq \lceil \frac{O(d \log d)}{\epsilon} \log \frac{O(d \log d)}{\epsilon} \rceil$. Now, let $\epsilon = cdn^{-1/d} \log n$, where $c$ is some constant. Then, there is an $\epsilon$-net of $Q$ of size at most

$$O(d \log d) \cdot \frac{n^{1/d}}{cd \log n} \log \left( O(d \log d) \frac{n^{1/d}}{cd \log n} \right) \leq c_1 n^{1/d}$$

where $c_1 \leq O(1)$. For large enough $c$, we can get small enough $c_1$ such that $\pi^*(\lfloor c_1 n^{1/d} \rfloor) < n$. For some constant $c$, we thereby get a $(cdn^{-1/d} \log n)$-net of $Q$ for the ranges $R_{xy}$ with $\pi^*(|N|) < n$. This implies that there is a pair of elements $\{x, y\}$ crossed by more than $cd|Q|n^{-1/d} \log n = cdmn^{-1/d} \log n$ ranges in $Q$. Finally, applying $d \leq \log n$ concludes the argument. $\qquad\square$

**Lemma 3.14.** *For a range space $(X, R)$, $|X| = n$ with spanning tree $T$, there exists a permutation $\pi \in S_n$ such that the weight of the corresponding spanning tree $T'$ (which is just the line $\pi(1), \pi(2), ..., \pi(n)$) is at most twice that of $T$.*

*Proof.* Enumerate all the ranges in $R$ by $R_1, \ldots, R_m$, where $R_i := \{c : \mathbf{M}_{i,c} = 1\}$ for $i \in [m]$.

The crossing number of $R_i$ in $T$ counts the number of edges in $T$ such that one endpoint of the edge is in $R_i$ and the other endpoint is in $[n] \setminus R_i$. Suppose the spanning tree $T$ on $X$ has a crossing number of $\kappa_i$ for just $R_i$ and let $w(T) = \sum_{i=1}^{m} \kappa_i$ be the weight of $T$ for $(X, R)$.

Perform a DFS traversal on $T$ and write the in-order traversal $\mathcal{I}$. We use the permutation $\pi(i) = \mathcal{I}_i$ for $i \in [n]$. Let $T'$ be the corresponding linear tree. In the worst case, this vertex path crosses $R_i$ $2\kappa_i$ times, that is because DFS uses each edge $(x, y)$ of $T$ twice: once when visiting $y$ (going down one level in the tree) and once when it is done processing $y$ (DFS pops the current level from the stack).

Thus the total number of crossings $w(T')$ for the linear tree $T'$ is $w(T') \leq \sum_{i=1}^{m} 2\kappa_i \leq 2\kappa(T)$. $\square$

## B.3 Dynamic OMv Data Structure

In this section we give a black-box reduction from dynamic OMv to static OMv. In the static version, we are given a matrix $\mathbf{M}$ to preprocess. This matrix stays fixed, while we receive an online sequence of vectors and must repeatedly compute $\mathbf{M}v$. In the dynamic version, we can inform the data structure about changes to $\mathbf{M}$. Specifically, we are allowed to add/remove/change rows/columns of the matrix.

Using the reduction from dynamic to static, we obtain Theorem 1.4.

**Theorem 1.4.** *(Dynamic Online Matrix-Vector Multiplication). Given a matrix $\mathbf{M} \in \{0, 1\}^{m \times n}$, there is a data structure $\mathcal{D}$ with $\widetilde{O}(mn)$ preprocessing time that supports row and column updates (insertions/deletions) to $\mathbf{M}$ in $\widetilde{O}(n)$ and $\widetilde{O}(m)$ time, respectively. Upon querying $\mathcal{D}$ with a vector $v \in \mathbb{R}^n$, it outputs $\mathbf{M}v$ in $\widetilde{O}(nm^{1-1/d^*} + m)$ time, with high probability, where $d^*$ is the largest corrupted VC-dimension of $\mathbf{M}$ throughout the history of its updates.*

The idea of this reduction is to split the columns of $\mathbf{M}$ into $\log n$ matrices $\mathbf{M}_0, ..., \mathbf{M}_{\log n}$ where $\mathbf{M}_i$ contains at most $2^i$ columns. Each $\mathbf{M}_i$ is used as input to a static OMv data structure which is reinitialized whenever $\mathbf{M}_i$ changes which occurs at most once every $2^i$ updates to $\mathbf{M}$. This allows to amortize the reinitialization cost over multiple updates.

**Lemma B.7.** *Assume there is a static OMv data structure for matrices of size $m \times n$ with preprocessing time $P(m, n)$ and query time $Q(m, n)$. Then there exists a dynamic OMv data structure supporting row and column insertions and deletions. The preprocessing time is $\widetilde{O}(P(m, n))$, the amortized row update time is*

$$O\left( \sum_{j=0}^{\log m} \frac{P(2^j, n)}{2^j} \right)$$

*and for column updates*

$$O\left( \sum_{i=0}^{\log n} \frac{P(m, 2^i)}{2^j} \right)$$

*and query time is $\widetilde{O}(Q(m, n))$. This reduction runs several static OMv data structures, each receiving a submatrix of the dynamic input matrix.*

We start by proving column updates only, then extend the reduction to support row updates as well.

**Lemma B.8.** *Assume there is a static OMv data structure for matrices of size $m \times n$ with preprocessing time $P(m, n)$ and query time $Q(m, n)$. Then there exists a dynamic OMv data structure supporting column insertions and deletions. The preprocessing time is $\widetilde{O}(P(m, n))$, the query time is $\widetilde{O}(Q(m, n))$, and the amortized update time is*

$$O\left( \sum_{i=0}^{\log m} \frac{P(m, 2^i)}{2^i} \right)$$

*This reduction runs several static OMv data structures, each receiving a submatrix of the dynamic input matrix.*

*Proof.* Let $\mathbf{M}$ be the input matrix that changes over time.

**Initialization.** We run $\log n$ instances of the static OMv data structure, where the $i$th instance is guaranteed to run on a matrix with at most $2^i$ columns. Let $\mathbf{M}^{(0)}, ..., \mathbf{M}^{(\log n)}$ be the respective input matrices to these instances. Initially, only the last ($i = \log n$) data structure receives the initial matrix $\mathbf{M}$ as input, and all other instances of the data structure initialize on an empty matrix. Hence initialization takes $O(P(m, n))$ time.

**Insertions.** Whenever there is a column insertion to $\mathbf{M}^{(i)}$ we reinitialize the $i$th static OMv data structure on that new matrix. When $\mathbf{M}^{(i)}$ has more than $2^i$ columns, then all columns are removed (i.e., we set $\mathbf{M}^{(i)} = 0$) and inserted to matrix $\mathbf{M}^{(i+1)}$. All column insertions to $\mathbf{M}$ are passed to $\mathbf{M}^{(0)}$. Observe that any $\mathbf{M}^{(i)}$ is updated every $2^i$ updates to $\mathbf{M}$, hence the amortized update time is

$$O\left(\sum_{i=0}^{\log n} P(m, 2^i)/2^i\right).$$

**Queries.** We can answer queries by observing that
$$\mathbf{M} = [\mathbf{M}^{(\log m)}|...|\mathbf{M}^{(1)}|\mathbf{M}^{(0)}].$$
So, for a query vector $v$ we split it into corresponding pieces $v^{(0)}, ... v^{(\log m)}$ and query each $\mathbf{M}^{(i)} v^{(i)}$. The time for this is
$$O(Q(m, n) \log n)$$
because each $\mathbf{M}^{(i)}$ is a submatrix of $\mathbf{M}$ and we have $\log n$ such matrices.

**Deletions.** When deleting a column of $\mathbf{M}$, we do not delete the column from the respective $\mathbf{M}^{(i)}$. Instead, all future query vectors $v$ will have an extra 0 entry for the deleted column.

When all columns from $\mathbf{M}^{(i)}$ are appended to $\mathbf{M}^{(i+1)}$ because of some insertions, then the deleted columns are not appended to $\mathbf{M}^{(i+1)}$. Hence whenever a static OMv data structure is initialized, it is a submatrix of the input matrix $\mathbf{M}$.

When there have been $2^i/2$ postponed deletions to any $\mathbf{M}^{(i)}$, then the columns are actually deleted and the respective static OMv data structure is reinitialized on the new $\mathbf{M}^{(i)}$. The remaining columns in $\mathbf{M}^{(i)}$ all exist in the current dynamic input $\mathbf{M}$, so again, the static OMv data structure receives a submatrix of $\mathbf{M}$ as input. For any one $i$, this takes amortized $O(P(n, 2^i)/2^i)$ time which is subsumed by the complexity of handling insertions.

Observe that internally, our matrices $\mathbf{M}^{(i)}$ are always at most twice their intended size because of the yet to be removed columns. Thus the query complexity increases by at most a constant factor. □

We now prove Lemma B.7 (Row & Column Updates)

*Proof.* Here, we extend Lemma B.8 to support row updates as well. The reduction follows the same idea as Lemma B.8.

Let $\mathbf{N}$ be the dynamic input matrix. We run $\log n$ copies of the column update data structure Lemma B.8 on matrices $\mathbf{N}^{(0)}, ..., \mathbf{N}^{(\log n)}$, with the $i$'th matrix having at most $2^i$ rows. When inserting new rows to $\mathbf{N}$, they are passed to $\mathbf{N}^{(0)}$. When $\mathbf{N}^{(i)}$ has more than $2^i$ rows, all rows are appended to $\mathbf{N}^{(i+1)}$. When rows are appended to any $\mathbf{N}^{(i)}$, the respective data structure is reinitialized. To handle deletions, we remove the respective row from the output until $2^i/2$ rows have been deleted from $\mathbf{N}^{(i)}$, then the rows are removed from $\mathbf{N}^{(i)}$ and the respective data structure is reinitialized. The amortized row update time thus becomes

$$O\left(\sum_{j=0}^{\log n} \frac{\text{Time to initialize Lemma B.8 on } 2^j \times n \text{ matrix}}{2^j}\right) = O\left(\sum_{j=0}^{\log m} \frac{P(2^j, n)}{2^j}\right).$$

For column updates, each of the $\mathbf{N}^{(i)}$ receives a new column, so each of the $\log m$ column update data structures is updated which takes:

$$O\left(\sum_{j=0}^{\log m} \sum_{i=0}^{\log n} \frac{P(2^j, 2^i)}{2^i}\right) = O\left(\sum_{i=0}^{\log n} \frac{P(m, 2^i)}{2^i}\right)$$

because geometric sums are bounded by their largest term in $O$-notation.

Finally, the query time is

$$\widetilde{O}\left(\sum_{j=0}^{\log m} Q(2^j, n)\right) = \widetilde{O}(Q(m, n)),$$

which proves the lemma. $\qquad\square$

We now prove Theorem 1.4 (Dynamic OMv for corrupted VC-dimension $d$).

*Proof.* The initialization cost of static OMv is $P(m, n) = \widetilde{O}(mn)$, and the query complexity is $Q(m, n) = \widetilde{O}(nm^{1-1/d} + m)$ by Theorem 1.3. This complexity also holds for submatrices, i.e., $Q(a, b) = \widetilde{O}(nm^{2-1/d} + m)$ for any $a \times b$ sized submatrix.

Thus we have update time for column updates:

$$O\left(\sum_{i=0}^{\log n} \frac{P(m, 2^i)}{2^i}\right) = \widetilde{O}\left(\sum_{i=0}^{\log n} \frac{m2^i}{2^i}\right) = \widetilde{O}(m)$$

and for row updates we have

$$O\left(\sum_{j=0}^{\log m} \frac{P(2^j, n)}{2^j}\right) = \widetilde{O}\left(\sum_{j=0}^{\log m} \frac{2^j n}{2^j}\right) = \widetilde{O}(n)$$

and for query time we have

$$\widetilde{O}(Q(m, n)) \leq \widetilde{O}(nm^{1-\frac{1}{d}} + m),$$

which proves the theorem. $\qquad\square$

## B.4  Extension to Structured Non-Boolean Matrices

In this section, we prove a result for a subquadratic-time accelerated matrix-vector multiplication for non-Boolean matrices, when the analogous *Pollard pseudodimension* is bounded. We also list further applications of this result.

**Theorem B.9.** *Suppose that the Pollard pseudodimension of the matrix $\mathbf{M} \in \mathbb{R}^{m \times n}$ is $d$, and that the number of distinct values per column is $\leq T$. Then, after an $\widetilde{O}(Tmn)$ time preprocessing, there is a data structure that, upon receiving a vector $v \in \mathbb{R}^n$, returns $\mathbf{M}v$ in time $O(Tnm^{1-1/d} + m)$.*

**Remark B.10.** *When $\mathbf{M}$ is the weight matrix of a neural network and $v$ is the activation output from the previous layer of the neural network, the matrix-vector product $\mathbf{M}v$ corresponds to running inference on $v$ with the neural network $\mathbf{M}$. This is a key step in a number of applications, including online optimization in unknown no-linear systems [Lin et al., 2023b] and inference in large language models. Dehghankar et al. [2024] studies the latter application with ternary matrices $\mathbf{M} \in \{-1, 0, 1\}^{n \times m}$ and shows that they are amenable to $O(n^2/\log n)$ multiplication. If $\mathbf{M}$ has bounded Pollard pseudodimension $d$, then as $T = 3$, we obtain a polynomial acceleration on this result: by Theorem B.9, the matrix-vector product can be computed in $\widetilde{O}(mn^{1-1/d} + n)$ time.*

Recall the definition of the Pollard pseudodimension:

**Definition B.11** (Pollard Pseudodimension). *For a family of functions $\mathcal{F}$ mapping $X$ to $Y$, the Pollard pseudodimension of $\mathcal{F}$, denoted by $\mathrm{Pdim}(\mathcal{F})$, is the VC dimension of the set system $(X \times Y, \{\{(x, y) \mid f(x) \geq y\} \mid f \in \mathcal{F}\})$.*

In the setting of matrices $\mathbf{M} \in \mathbb{R}^{m \times n}$, we interpret each row as a function $f_{\mathbf{M},i} : x \mapsto \mathbf{M}_{i,x}$. In particular, when $\mathbf{M}$ has Pollard pseudodimension $d$, then that means the set system $([n] \times \mathbb{R}, \{(j, y) \mid \mathbf{M}_{i,j} \geq y\} \mid i \in [m]\})$ has VC-dimension $d$. Restricting the groundset $[n] \times \mathbb{R}$ to $\{(j, y) \mid \exists i : \mathbf{M}_{i,j} = y\}$ does not increase the VC-dimension. This latter set system can be interpreted as the following Boolean matrix $\mathbf{N}$:

Given a matrix $\mathbf{M} \in \mathbb{R}^{m \times n}$ where each column has at most $T$ distinct values, consider the following construction of an $\mathbf{N} \in \{0, 1\}^{m \times nT}$ matrix. Let $\mathbf{Y} \in \mathbb{R}^{T \times n}$ be the distinct values per column of $\mathbf{M}$

(we may duplicate values if a column has $< T$ distinct values). Assume the entries in each column of $\mathbf{Y}$ are sorted in increasing order. Let $\mathbf{N}_{i,jT+k} = \mathbb{1}_{\mathbf{M}_{i,j} \geq \mathbf{Y}_{i,k}}$.

For example, the following $\mathbf{M}$ has at most $T = 4$ distinct values per column:

$$
\mathbf{M} := \begin{bmatrix} 1 & -1 \\ 4 & -2 \\ 2 & 4 \\ 5 & 3 \\ 4 & 3 \end{bmatrix} \rightarrow \left[\begin{array}{cccc|cccc} 1 & 0 & 0 & 0 & 1 & 0 & 0 & 0 \\ 1 & 1 & 0 & 0 & 1 & 1 & 0 & 0 \\ 1 & 1 & 1 & 1 & 1 & 1 & 1 & 1 \\ 1 & 1 & 1 & 0 & 1 & 1 & 1 & 0 \\ 1 & 1 & 0 & 0 & 1 & 1 & 1 & 0 \end{array}\right] =: \mathbf{N}, \quad \mathbf{Y} := \begin{bmatrix} 1 & -2 \\ 2 & -1 \\ 4 & 3 \\ 5 & 4 \end{bmatrix}
$$

Since matrix $\mathbf{N}$ is Boolean and of VC-dimension at most $d$, we can run our fast matrix-vector product data structures on it (e.g., Theorem 1.3).

*Proof of Theorem B.9.* Given $\mathbf{M} \in \mathbb{R}^{m \times n}$ we construct the Boolean matrix $\mathbf{N} \in \{0,1\}^{m \times nT}$ as previously described.

We initialize Theorem 1.3 on $\mathbf{N}$ which takes $\widetilde{O}(Tmn)$ time. Multiplying a vector $v' \in \mathbb{R}^{nT}$ with $\mathbf{N}$ takes $\widetilde{O}(Tnm^{1-1/d} + m)$ time. We are left with explaining how to compute $\mathbf{M}v$ via $\mathbf{N}v'$.

We have

$$
\mathbf{M}_{i,j} = \mathbf{Y}_{1,j} + \sum_{k>1: \mathbf{Y}_{k,j} \leq \mathbf{M}_{i,j}} \mathbf{Y}_{k,j} - \mathbf{Y}_{k-1,j}
$$

$$
= \mathbf{N}_{i,jT} \mathbf{Y}_{1,j} + \sum_{k=2}^{T} \mathbf{N}_{i,j \cdot T+k}(\mathbf{Y}_{k,j} - \mathbf{Y}_{k-1,j})
$$

In particular, we can compute $\mathbf{M}v = \mathbf{N}v'$ where

$$
v'_{j \cdot T} = v_j \cdot \mathbf{Y}_{1,j}, \ v'_{j \cdot T+k} = v_j \cdot (\mathbf{Y}_{k,j} - \mathbf{Y}_{k-1,j}) \text{ for } k \geq 2.
$$

Constructing $v'$ takes $O(Tn)$ time per query. $\qquad\square$

Matrices with small constant Pollard pseudodimensions are also prevalent in real-world data, and capture a similar notion of structure as matrices with low constant VC-dimensions. For instance, the following are examples of matrices with small Pollard pseudodimension.

**Definition B.12** (Multiball vectors). *Let $v \in V$ be a vertex of the graph $G$. Let $r$ be a real number and $\Delta$ be a set of distances $-\infty = \delta_0 < \delta_1 < \cdots < \delta_{\ell-1} < \delta_\ell = \infty$. Define the multiball vectors as follows:*

$$
\vec{\mathrm{MB}}(v, r, \Delta) = (y_u)_{u \in V} \text{ where } y_u \in [\ell] \text{ is the smallest integer such that } u \in \vec{B}(v, r + \delta_{y_u}).
$$

*Let $\vec{\mathrm{MB}}_{G,\Delta} = \{\vec{MB}(v,r,\Delta) | v \in V, r \in \mathbb{R}\}$ be the set of all multiball vectors in $G$ with shifts $\Delta$.*

**Theorem B.13** (Karczmarz and Zheng [2024]). *For a $K_h$-minor-free graph $G$, for any set $\Delta$ of $\ell$ distances, the set of multiball vectors $\vec{\mathrm{MB}}_{G,\Delta}$ has pseudodimension at most $h - 1$.*

## C  Applications

In this section, we consider some applications of our OMv result in Theorem 1.4. These applications have quadratic $\Omega(n^2)$-time lower bounds, conditional on the OMv conjecture. We show that on structured graph this lower bound can be beaten.

### C.1  High-accuracy Dynamic Laplacian Solver

**Theorem 1.6.** *(Dynamic Laplacian Solver). There is a dynamic algorithm that, given a dynamic graph $G = (V, E)$ with corrupted VC-dimension bounded by $d$, maintains a Laplacian system solver. The data structure supports queries that receive a vector $b \in \mathbb{R}^{|V|}$ and error parameter $\epsilon > 0$. Then, in $\widetilde{O}(n^{2-1/d} \log 1/\epsilon)$ time, the algorithm returns the (approximate) solution $x$ to $\mathbf{L}x^* = b$ where $\|x - x^*\|_{\mathbf{L}} \leq \epsilon \|x^*\|_{\mathbf{L}}$. Each vertex update to $G$ takes $\widetilde{O}(n)$ time.*

We consider the setting where the Laplacian $\mathbf{L}$ corresponds to a graph with bounded VC-dimension. First, recall the notion of spectral sparsifiers.

**Definition C.1** $((1+\epsilon)$-spectral sparsifiers). *Let $\mathbf{L}_X$ denote the Laplacian of any undirected graph $X$. Then, a $(1 \pm \epsilon)$-spectral sparsifier $H$ of a graph $G$ is a subgraph of $G$ such that for every vector $x \in \mathbb{R}^n$,*

$$(1-\epsilon)x^\top \mathbf{L}_H x \leq x^\top \mathbf{L}_G x \leq (1+\epsilon)x^\top \mathbf{L}_H x,$$

Then, a result from Abraham et al. [2016] provides a construction for a dynamic spectral sparsifier under edge deletions and insertions with polylogarithmic amortized update time:

**Theorem C.2** (Theorem 4.1 of Abraham et al. [2016]). *There exists a fully dynamic randomized algorithm with polylogarithmic update time for maintaining a $(1 \pm \epsilon)$-spectral sparsifier $H$ of a graph $G$, with probability at least $1 - 1/n^c$ for any $0 < \epsilon \leq 1$ and $c \geq 1$. Specifically, the amortized update time of the algorithm is $O(c\epsilon^{-2} \log^3 \rho \log^6 n)$ and the size of $H$ is $O(cn\epsilon^{-2} \log^3 \rho \log^5 n \log W + m\rho^{-1})$, where $1 \leq \rho \leq m$ is a parameter of choice. Here, $W$ is the ratio between the largest and the smallest edge weight in $G$.*

We use the following result for solving Laplacian systems in the static setting.

**Lemma C.3** (Kyng and Sachdeva [2016]). *There is a randomized procedure that given any $n$-vertex $m$-edge graph $G$ with Laplacian matrix $\mathbf{L}_G$, and vector $b \in \mathbb{R}^V$ such that there exists an $x \in \mathbb{R}^V$ with $\mathbf{L}_G x = b$ computes $\overline{x} \in \mathbb{R}^V$ with $\|\overline{x} - x\|_{\mathbf{L}_G} \leq \epsilon\|x\|_{\mathbf{L}_G}$ in $\widetilde{O}(m \log \epsilon^{-1})$ with high probability.*

**Lemma C.4.** *Let $G$ be a dynamic graph and let $\mathbf{A}$ be its dynamic adjacency matrix. Assume there is a dynamic OMv data structure for this $\mathbf{A}$ with update time $U(n)$ and query time $Q(n)$.*

*Then there exists a dynamic Laplacian System solver for the same dynamic graph $G$, supporting updates to $G$ in $O(U(n) + n)$ time. The data structure supports queries which for any given $b \in \mathbb{R}^V$ and $\epsilon > 0$ in $\widetilde{O}(Q(n) \log 1/\epsilon)$ time return $x \in \mathbb{R}^V$ with $\|x - x^*\|_{\mathbf{L}_G} \leq \epsilon\|x^*\|_{\mathbf{L}_G}$ where $x^*$ is the exact solution for $\mathbf{L}_G x = b$.*

Via Theorem 1.4 we have $U(n) = \widetilde{O}(n)$ and $Q(n) = \widetilde{O}(n^{2-1/d})$, thus obtaining Theorem 1.6.

*Proof.* We run two separate data structures. The first is a dynamic spectral sparsifier Theorem C.2 which maintains a spectral sparsifier $H \approx \mathbf{L}_G$ with error $\epsilon = 1/2$. We also run the dynamic OMv data structure for matrix $\mathbf{A}$ (adjacency matrix).

The update time is thus $\widetilde{O}(n + U(n))$ as the spectral sparsifier needs polylog time per edge updates, thus $\widetilde{O}(n)$ time for node updates.

**Queries** When given a vector $b \in \mathbb{R}^V$ and error parameter $\epsilon > 0$, we use $\mathbf{H}^{-1}$ as preconditioner for the linear system $\mathbf{L}_G x = b$. We can multiply with $\mathbf{H}^{-1}$ by running a static Laplacian system solver Lemma C.3 which takes $\widetilde{O}(n)$ time per product as $\mathbf{H}$ has $\widetilde{O}(n)$ edges. In particular, we let

$$x^0 = \mathbf{H}^{-1}b$$

and then perform iterative refinement (Richardson iteration)

$$x^{t+1} \leftarrow x^t - \mathbf{H}^{-1}(\mathbf{L}_G x^t - b).$$

After $\widetilde{O}(\log 1/\epsilon)$ iterations we have $\|x^t - x^*\|_{\mathbf{L}_G} \leq \epsilon\|x^*\|_{\mathbf{L}_G}$.

Each iteration takes $\widetilde{O}(n + Q(n))$ time, where $\widetilde{O}(n)$ is the time for multiplying by $\mathbf{H}^{-1}$, and $Q(n)$ the time for multiplying by $\mathbf{L}_G$. Observe that $\mathbf{L}_G v = \mathbf{D}v - \mathbf{A}v$ where $\mathbf{D}$ is a diagonal matrix and $\mathbf{A}$ is an adjacency matrix. So the first product takes $O(n)$ time and the second product is handled by the dynamic OMv data structure in $O(Q(n))$ time. Together, this proves the lemma.

$\square$

## C.2 Effective Resistance

The following Theorem 1.7 is a corollary of the dynamic Laplacian solver from Theorem 1.6.

**Theorem 1.7.** *(Dynamic Effective Resistance). There is a dynamic algorithm that, given a dynamic graph $G = (V, E)$ with corrupted VC-dimension bounded by $d$, maintains effective resistances in $G$. The data structure supports queries that receive a pair of vertices $u, v \in V$ and error parameter $\epsilon > 0$. Then, in $\widetilde{O}(n^{2-1/d} \log 1/\epsilon)$ time, the algorithm returns a $(1 \pm \epsilon)$-approximation of the effective resistance. Moreover, each node update to $G$ in the dynamic data structure takes $\widetilde{O}(n)$ time.*

*Proof.* The effective resistance of a pair $u, v \in V$ is the energy of an electric flow, routing one unit of flow between $u$ and $v$. The energy of an electric flow $f \in \mathbb{R}^E$ is $\sum_e f_e^2$ and the electric flow is given by $f = \mathbf{B}\mathbf{L}^\dagger(e_u - e_v)$ where $\mathbf{B} \in \{-1, 0, 1\}^{E \times V}$ is the incidence matrix of $G$ with arbitrary directions. Thus the effective resistance is

$$\|\mathbf{B}\mathbf{L}^\dagger(e_u - e_v)\|_2^2 = (e_u - e_v)^\top \mathbf{L}^\dagger \mathbf{B}^\top \mathbf{B}\mathbf{L}^\dagger(e_u - e_v)$$
$$= (e_u - e_v)^\top \mathbf{L}^\dagger(e_u - e_v).$$

This can be computed via the dynamic Laplacian system solver in Theorem 1.6. □

## C.3 Triangle Detection

**Theorem 1.8.** *(Dynamic Triangle Detection). There is an algorithm that, given a dynamic graph $G = (V, E)$ with corrupted VC-dimension $d$, maintains whether $G$ has a triangle or not. Each vertex update takes $\widetilde{O}(n^{2-1/d})$ time and returns a Boolean indicator for $G$ containing a triangle.*

The result follows form the following lemma for $U(n) = \widetilde{O}(n)$ and $Q(n) = \widetilde{O}(n^{2-1/d})$ from Theorem 1.4.

**Lemma C.5.** *Let $G$ be a dynamic graph and let $\mathbf{A}$ be its dynamic adjacency matrix. Assume there is a dynamic OMv data structure for this $\mathbf{A}$ with update time $U(n)$ and query time $Q(n)$.*

*Then there exists a dynamic triangle detection algorithm for the same dynamic graph $G$, supporting updates in $O(U(n) + Q(n))$ time.*

*Proof.* Let $\mathbf{A}$ be the adjacency matrix of the graph, then the diagonal entries $(\mathbf{A}^3)_{v,v}$ are non-zero if and only if $v$ participates in a triangle. Thus by maintaining the sum $\sum_i (\mathbf{A}^3)_i$ we can detect if there is any triangle in the graph.

We maintain the sum as follows: Let $\mathbf{A}$ be the incidence matrix before, and $\mathbf{A}$ after a node $v$ is updated.

Then

$$\sum_i (\mathbf{A}'^3)_i = (\sum_i (\mathbf{A}^3)_i) + 3(\underbrace{(\mathbf{A}'^3)_{v,v} - (\mathbf{A}^3)_{v,v}}_{\text{difference in \# of triangles } v \text{ participates in}}).$$

This is because each triangle $v$ participates in is counted thrice in the sum (once for each vertex of the triangle). Thus we can maintain the sum by computing $(\mathbf{A}'^3)_{v,v}$ and $(\mathbf{A}^3)_{v,v}$. Observe that this is the vector-matrix-vector product of (i) the $v$th row of $\mathbf{A}$, (ii) matrix $\mathbf{A}$, (iii) and $v$-th column vector of $\mathbf{A}$. Thus it can be computed in $O(Q(n))$ time, with an extra $U(n)$ time to update $\mathbf{A}$ to $\mathbf{A}'$. □

## C.4 Single-Source Shortest Paths and k-Center

**Lemma C.6** (Brand et al. [2022]). *Assume there is a dynamic distance oracle for unweighted undirected graphs with the following operations.*

*An initialization procedure that is given $G = (V, E)$ and a threshold $1 \le d \le n$.*

*A node update operation with update time $U(n, d)$.*

*A query operation which for any source node, given during the query, returns the $d$-bounded single source distances in $O(Q(n, d))$. That is, return for each $v \in V$ the distance if it is at most $d$, or $\infty$ if the distance is $> d$.*

*Then for any $1 \leq k \leq n$ and $\epsilon > 0$ there exists a dynamic $(1+\epsilon)$-approximate single source distance data structure on unweighted undirected graphs. It supports node updates in*

$$O\left(U\left(n, \frac{4}{\epsilon}\right) + \frac{k}{\epsilon} \cdot Q\left(n, \frac{4}{\epsilon}\right) + \frac{n^2}{k}\right)$$

*time.*

**Theorem 1.9.** *(Dynamic Approximate Single-Source Shortest Paths). There is a dynamic algorithm that maintains $(1+\epsilon)$-approximate single-source distances on a dynamic unweighted graph $G = (V, E)$. If the corrupted VC-dimension of $G$ is bounded by $d$, each node update to $G$ takes $\widetilde{O}(kn^{2-1/2d}/\epsilon)$ time, and querying the distances for any source node takes $\widetilde{O}(n^{2-1/2d}/\epsilon)$ time.*

*Proof.* We first describe how to obtain a dynamic distance oracle that yields distances up to $1/\epsilon$. This is done by running the dynamic OMv data structure of Theorem 1.4 on the adjacency matrix of $G$. Then single source distances up to $1/\epsilon$ from source vertex $u$ can be computed by repeatedly multiplying $w \leftarrow \mathbf{A}w$ for initial $w = e_u$, i.e., we compute $\mathbf{A}^i e_u$ for $i = 1, 2, ..., 1/\epsilon$. The smallest $i$ with $(\mathbf{A}^i e_u)_v \neq 0$ is the smallest number of steps to reach from $u$ to $v$, i.e., the distance between the vertices.

This data structure internally constructs the a $\Delta$-labeled spanning tree of the boolean matrix $\mathbf{A}$, where the weight of the tree gives us the time complexity per query. Hence, while we do not know the corrupted VC-dimension $d$, we do know how much time each query is going to take.

So when running Lemma C.6 on the above distance data structure, we can pick $k = n/\sqrt{T}$ where $T \geq n$ is the weight of the spanning tree. Thus, we get the update time

$$\widetilde{O}\left(n + \frac{k}{\epsilon}T + nk + \frac{n^2}{k}\right) = \widetilde{O}\left(n + \frac{n\sqrt{T}}{\epsilon} + \frac{n}{\sqrt{T}}\right)$$
$$= \widetilde{O}(n^{2-\frac{1}{2d}}),$$

which proves the claim. $\qquad\square$

**Theorem 1.10.** *(Dynamic Approximate $k$-center). Given an unweighted undirected graph $G = (V, E)$, there is a dynamic algorithm for $(2 + \epsilon)$-approximate $k$-center with node update time $\widetilde{O}(kn^{2-1/2d}/\epsilon)$, where $d$ is a bound on the corrupted VC-dimension of $G$.*

This result is a corollary of Theorem 1.9, as $k$-center can be reduced to computing $k$-single source distances.

**Lemma C.7** (Theorem 7.3 by Cruciani et al. [2024]). *Given a graph $G = (V, E)$, a positive parameter $\epsilon \leq 1/2$, and a fully-dynamic data structure that maintains $(1+\epsilon)$-approximate single source distances with update time $T(n, m, \epsilon)$ and $Q(n, m, \epsilon)$ query time, there is a dynamic algorithm that maintains a $2(1 + 4\epsilon)$-approximate solution to fully-dynamic $k$-center in time $O(T(n, m, \epsilon) + k \cdot (Q(n, m, \epsilon) + n))$.*

## D   Characterization of matrices with constant VC-dimension

In this section, we provide a characterization of matrices with constant VC-dimension. Particularly, we prove Theorem 1.1 and state a number of examples of other constant VC-dimension matrices.

**Fact 1.** *Consider a hereditary class of $0/1$-matrices $\mathcal{M}$, meaning that $\mathcal{M}$ is closed under row/column deletion (i.e., each matrix $\mathbf{M} \in \mathcal{M}$ is still in $\mathcal{M}$ after deleting a row or column). If $\mathcal{M}$ is non-trivial, (i.e., does not contain every possible matrix), then there exists an absolute constant $c \in \mathbb{N}$ such that the VC-dimension of any matrix in $\mathcal{M}$ is at most $c$.*

Fact 1.1 provides a structural characterization of some matrices with constant VC-dimension, and it is an analogous result to the fact that non-trivial hereditary classes of graphs have low VC-dimension. To prove Fact 1.1, we introduce the notion of a bipartisation of a graph.

**Definition D.1.** *A bipartite graph $H = (A \cup B, E)$ is a bipartisation of graph $G$ if removing all edges in $A$ and $B$ in $G$ yields $H$ for some partition $A, B$ of $V(G)$. A graph class $\mathcal{G}$ is said to be hereditary if it is closed under induced subgraphs.*

**Lemma D.2** (Lemma 3.4 of Bousquet et al. [2015])**.** *For any hereditary class $\mathcal{C}$ of graphs and for any bipartite graph $H$, if the graphs in $\mathcal{C}$ have infinite VC-dimension, then $\mathcal{C}$ contains a graph $G$ whose bipartisation is $H$.*

By transposition, we get the following corollary:

**Corollary D.3.** *The equivalent contrapositive statement is that for any hereditary class $\mathcal{C}$ of graphs and for any bipartite graph $H$, if $C$ does not contain a graph $G$ whose bipartisation is $H = (A \cup B, E)$ where $A \cup B = V(G)$, then the graphs in $C$ have finite VC-dimension.*

**Lemma D.4.** *The VC-dimension of any matrix in $\mathcal{M}$ is upper bounded by a finite constant.*

*Proof.* Since $\mathbf{M} \in \mathcal{M}$ is Boolean, we can treat it as a bipartite graph $G$, with rows representing left vertices and columns representing right vertices. In particular, we can treat $\mathcal{M}$ as a class of bipartite graphs closed under vertex deletion (row deletion = deleting a left vertex, column deletion = deleting a right vertex). Thus we can apply Lemma D.2 to $\mathcal{M}$.

Since $\mathcal{M}$ is non-trivial, there must be some graph $H$ not contained in it. For this excluded graph $H$, define $H'$ as the same graph but add one extra left vertex and one extra right vertex. Connect this new left vertex to every vertex on the right, and connect the new right vertex to every vertex on the left. Note that since $H$ was not contained in $\mathcal{M}$ and it is closed under vertex deletion, $H'$ also cannot be contained in $\mathcal{M}$.

Now assume there is $G \in \mathcal{M}$ that contains $H'$ as bipartization. Then there is a partition $A, B$ of $V$, such that keeping only the edges between $A$ and $B$ of $G$, results in $H'$. Because of the left vertex in $H'$ that connects to all right vertices, and the right vertex connecting to all left vertices, we must have that $A$ are all left and $B$ are all the right vertices of $G$ (or the other way around). But that means no edges were removed from $G$ when restricting to edges between $A$ and $B$, since a bipartite graph by definition only has edges connecting left and right vertices. So $G = H'$ and $H' \in \mathcal{M}$. This is a contradiction because by construction of $H'$, it does not exist in $\mathcal{M}$.

We conclude, that set of bipartite graphs corresponding to $\mathcal{M}$ does not contain any graph whose bipartisation is $H'$. Hence by Corollary D.3 the VC-dimension must be a finite constant. Note that this is equivalent to the VC-dimension of the adjacency matrices being bounded, but $\mathcal{M}$ are not adjacency matrices. The adjacency matrices are of form

$$\begin{bmatrix} 0 & \mathbf{M} \\ \mathbf{M}^\top & 0 \end{bmatrix}$$

for matrices $\mathbf{M} \in \mathcal{M}$. We already argued that each adjacency matrix has constant VC-dimension, and since removing rows does not increase the VC-dimension (it is equivalent to deleting sets), this implies $\mathbf{M}$ also has constant VC-dimension. In conclusion, each matrix in $\mathcal{M}$ has constant VC-dimension. $\square$

### D.1 Examples of low VC-dimension matrices

We provide some broad families of matrices with constant VC dimension.

**Adjacency matrices of $H$-minor free graphs.** Eppstein [1995] showed that $H$-minor free directed graphs (where the minor can be obtained through vertex deletion, edge deletion, and edge contraction) has VC-dimension at most $|V(H)| - 1$. As an immediate consequence, since bipartite-graphs are triangle ($K_3$) free, the VC-dimension of any bipartite graph is 2. As a consequence, given any adjacency matrix $\mathbf{A}$ of a $H$-minor free graph $G$, for any $k \geq 1$, $\mathbf{A}^k$ is the adjacency matrix of a $H$-minor free graph $G'$ (since $\mathbf{A}^k$ corresponds to the reachability graph where an edge $(u, v) \in E(G')$ corresponds to the existence of a $k$-length walk between $u$ and $v$ in $G$). Therefore, the VC-dimension of $\mathbf{A}^k$ is also at most $|V(H)| - 1$.

**Adjacency matrices of interval graphs.** Interval graphs are the intersection graphs of a family of intervals on the real line. Then, since the hypothesis class of intervals on the real line $\mathcal{H} = \{\mathbb{1}_{a,b} : a \leq x \leq b\}$ has VC-dimension 2 (any two points can be shattered by choosing an interval that

includes one or both points, but no set of three points can always be shattered), every interval graph has VC-dimension at most 2.

**Boolean kernel matrices.** A matrix where each column $i$ (and each row) receives some label $\ell_i \in L$ from some (possibly infinite) set of possible labels $L$. Then, suppose each entry of $\mathbf{M}$ satisfies $\mathbf{M}_{i,j} = f(\ell_i, \ell_j)$ for some function $f : L \to \{0, 1\}$. For example, the labels could be points in $L = \mathbb{R}^d$ with $f$ being indicator that the distance is at most some threshold. If $f, L$ is fixed (i.e., independent of the number of rows/columns) then these graphs are closed under row/column deletion and thus have bounded VC-dimension by Theorem 1.1.

**Shortest-path structures.** Let $G$ be an undirected graph with non-negative edge weights and let $S$ be a subset of its shortest paths such that, for every pair $(u, v)$ of distinct vertices, $S$ contains exactly one shortest path between $u$ and $v$. Marie de Lima et al. [2023] defines a range space associated with $S$ and proves that its VC dimension is 2.

**Adjacency matrices of planar graphs.** Let $G$ be a planar graph. Then, there is a planar drawing of $G$, and deleting vertices and edges retains this property. Therefore, these graphs are closed under row/column deletion and thus have bounded VC-dimension by Theorem 1.1.

**Adjacency matrices of semi-algebraic graphs of bounded description complexity.** By the Milnor-Thom theorem in real algebraic geometry [Matoušek, 2002] and Assouad's theorem [Assouad, 1983], semi-algebraic graphs of bounded description complexity have constant VC-dimensions. In turn, this extends to adjacency matrices of string graphs where every two curves intersect.

**Practical matrices.** Coudert et al. [2024] computes the VC-dimension of families of graphs that arise in practice, from protein interaction networks to autonomous Internet systems. Even for such graphs with millions of nodes, the VC-dimension of the graphs are typically between 3 and 8.

# E   Numerical Simulations

The algorithm from Theorem 1.3 has previously been described in Björklund and Lingas [2001] and Alves et al. [2024]. The latter already experimentally verified the efficiency of the algorithm on real-world data sets and observed that it beats the naive quadratic time matrix vector multiplication. Our work proves theoretic guarantees for this speed-up parameterized by the (corrupted) VC-dimension. While the main contribution of this work is to have theoretic guarantees and explanation for why the algorithm is efficient in practice, we here complement the result with a few experiments[3]. As we give theoretic bounds parameterized by $d$, our experiments focus on the $d$-dependence too.

**Matrix Generation.** Since we want to study the complexity dependence on $d$, we fix the matrix dimension to $n = 2^{12} = 4096$ and run experiments for $d = 2, ..., 12$. The maximum $d$ is 12 because the VC-dimension $\leq \log(n) = 12$ so we do not need to consider $d > \log(n)$. The matrix size $n$ was picked due to execution time constraints. The $n \times n$ Binary matrices are generated by sampling i.i.d uniform $a^{(i)} \in [0, 1]^d, b^{(i)} \in [0, d/2]$ for $i = 1, ..., n$ and $x^{(j)} \in [0, 1]^d$ for $j = 1, ..., n$, and letting $\mathbf{M}_{i,j} = \mathbb{1}_{\{a^{(i)\top} x^{(j)} \geq b^{(i)}\}}$. The range for the $b^{(i)}$ was chosen so that neither $\mathbf{M}$ nor $1 - \mathbf{M}$ are sparse, which otherwise would allow for naive matrix multiplication to already beat $O(n^2)$ time. The set system of linear classifiers has VC-dimension $d$, so the corrupted VC-dimension of $\mathbf{M}$ is upper bounded by $d$. This implies that the respective MST has weight at most $\widetilde{O}(n^{2-1/d})$. Figure 1 shows the observed weight of the MST relative to the ambient dimension $d$. The MST was computed exactly since easier to implement and the preprocessing time complexity is not focus of this work.

**Complexity.** For the matrix vector products we generate i.i.d. uniform vectors $v \in [0, 1]^n$. For each $d$, we compute 1000 matrix vector products and measure the run time. Figure 2 shows the mean and standard deviation of the observed time. We also compare the time to the naive numpy matrix vector multiplication. Figure 2 shows that, since matrix size $n$ is fixed, the time complexity of numpy is constant for different $d$, whereas the MST-based algorithm is substantially faster for small $d$. As $d$ increases, the gap gets smaller as one would expect, given the $\widetilde{O}(n^{2-1/d})$ theoretic upper bound. The experiments were conducted on an Apple M4 Pro macOS model with 12 physical cores / threads, and 48 GB unified memory RAM, and an Apple Accelerate Framework BLAS backend. The total

---

[3]The code for the experiments is available at `https://github.com/emiletimothy/structural_complexity_of_matrix_vector_multiplication`

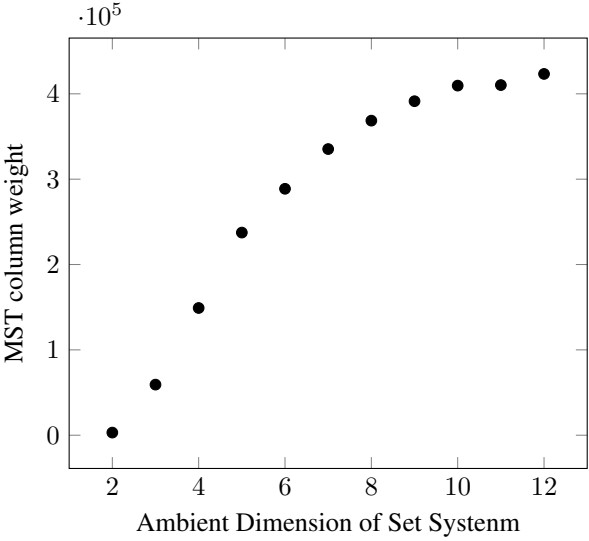

Figure 1: Weight of the $\Delta$-labeled spanning tree for $2^{12} \times 2^{12}$ matrices generated from halfspaces in $d$ dimensions.

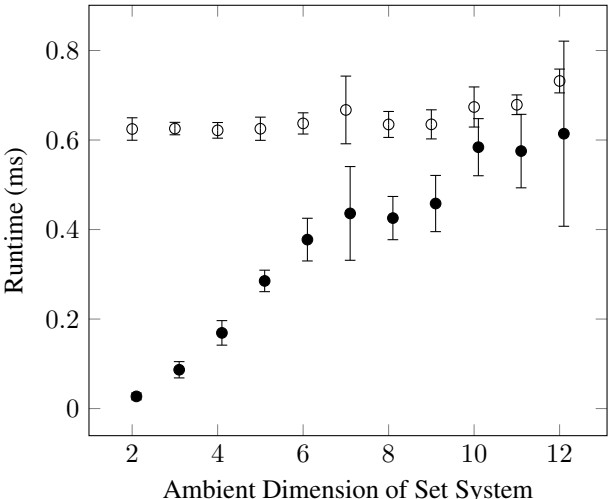

Figure 2: Average run time of the MST-based matrix vector algorithm (bold) and naive numpy matrix vector multiplication (blank), with 1-sigma error bars. Horizontal axis is the dimension of the points and halfspaces used to construct the matrix.

amount of time to run the experiments was $\approx 30$ minutes. No GPU acceleration was used; all matrix operations ran on CPU.

