# OpenReview forum: "The Structural Complexity of Matrix-Vector Multiplication"
_NeurIPS.cc/2025/Conference — NeurIPS 2025 poster_

### Official Review · Reviewer_goCC · 2025-06-25

**Clarity:** 3
**Significance:** 3
**Originality:** 3
**Rating:** 5
**Confidence:** 2

**Summary:**

This paper shows that for n x n Boolean matrices with VC dimension d (where the function class is defined by considering each row of the matrix as the evaluation table of a function), there is an O(n^{2-1/d}) time-per-query algorithm for online matrix-vector multiplication with preprocessing. It also presents various applications and extensions of this result.

**Questions:**

See weaknesses.

**Ethical Concerns:**

["NO or VERY MINOR ethics concerns only"]

**Final Justification:**

The authors' response addressed the minor issues that I found in the proofs. The result is an interesting theoretical contribution (if perhaps a bit niche) and I am happy to recommend acceptance.

**Limitations:**

yes

**Quality:**

3

**Strengths And Weaknesses:**

**Strengths.** The result gives a novel perspective on beyond-worst-case online matrix vector multiplication: VC dimension bounds are a plausibly natural assumption for real data, and this paper shows that they enable avoiding worst-case quadratic lower bounds. The basic result is an almost direct corollary of combining a (quite practical) algorithm due to (Bjorklund and Lingas, 2001) with an old (and to me unknown) lemma from the computational geometry literature showing that for any function class with bounded VC dimension, there are two points on which "most" of the functions agree (Welzl, 1988). Nevertheless making the connection between these two branches of literature is  a contribution in its own right, and the authors put substantial technical effort into extensions and applications of the basic result.

**Weaknesses.** I think there is a (fairly minor) gap in the current proof of the basic result: Welzl's Lemma 4.1 hides some "constants" that, I think, actually depend on the VC dimension bound d. The statement of Lemma 4.1 says "c is a constant depending on (X,R)", which is completely vacuous since (X,R) is the entire problem instance. From inspecting the proof, my impression is that c ~ O(d) suffices, but this should be checked and perhaps a self-contained modern proof can be included in the paper for completeness. At any rate, a factor of d is not really important since the result already requires d << log n for it to be non-trivial.

Also, I think Definitions 3.8 and 3.10 are wrong: in 3.8, the symmetric difference between two sets is non-empty iff the sets are non-equal. The second formulation of the definition of crossing is I think the correct one, but it's not the same as the first formulation. In 3.10, I think the crossing number should be "sum over edges of max over ranges of indicator that the range crosses the edge", not "max over ranges of sum over edges". I believe the correct version is used for the proofs though, so this seems to just be a typo.

I will increase my score if the authors can confirm that these are not serious issues.

---

> ### Author Rebuttal · Authors · 2025-07-30
>
> We thank the reviewer for their detailed comments, and for their constructive feedback to strengthen the quality of our paper. Please find our response below:
>
> > I think there is a (fairly minor) gap in the current proof of the basic result: Welzl's Lemma 4.1 hides some "constants" that, I think, actually depend on the VC dimension bound d. The statement of Lemma 4.1 says "c is a constant depending on (X,R)", which is completely vacuous since (X,R) is the entire problem instance. From inspecting the proof, my impression is that c ~ O(d) suffices, but this should be checked and perhaps a self-contained modern proof can be included in the paper for completeness. At any rate, a factor of d is not really important since the result already requires d << log n for it to be non-trivial.
>
> Thanks for raising this point. After checking Welzl’s proof of Lemma 4.1, we see that $c = O(d)$ indeed does suffice [1]. The proof of Lemma 4.1 in [1] should use $c_0 = O(d)$ because of the $8d$ factor in Proposition 2.4 of [1]. Parameter $c_1$ is chosen as $O(c_0/c)$ and needs to be $O(1)$ for the proof to go through. So it suffices to pick any $c$ bounded by $O(d)$.)
> We will add the respective updated proof to the appendix.
>
> This adds a $d$ factor to our time complexities, which (as the reviewer correctly points out) is at most a log-factor because $m\times n$ matrices can have VC-dimension at most $\log(m)$. However, since our $\tilde{O}$-notation hides such log-factors, this does not affect our algorithms and proofs. Therefore, our main result is unchanged.
>
> > Also, I think Definitions 3.8 and 3.10 are wrong: in 3.8, the symmetric difference between two sets is non-empty iff the sets are non-equal. The second formulation of the definition of crossing is I think the correct one, but it's not the same as the first formulation.
>
> Thank you for catching this error. Yes, crossing has nothing to do with symmetric difference, and we will remove that first formulation. However, as the reviewer correctly points out, we did in fact only use the second formulation of the definition of crossing ($A$ crosses $r$ if neither $A \subseteq r$ nor $A\cap r=\emptyset$ holds) in the proof of Theorem 3.7, so this does also not affect the main result.
>
> > In 3.10, I think the crossing number should be "sum over edges of max over ranges of indicator that the range crosses the edge", not "max over ranges of sum over edges". I believe the correct version is used for the proofs though, so this seems to just be a typo.
>
> Yes, that’s a typo. Thanks for pointing this out, and we will fix this typo.
>
> We thank the reviewer again for their valuable feedback. We hope our reply addresses any concerns in our work, and we would be happy to answer any further questions.
>
> Thanks,
>
> Authors
>
> ===========
>
> References
> [1] E. Welzl. Partition trees for triangle counting and other range searching problems

---

> > ### Comment · Reviewer_goCC · 2025-08-02
> >
> > Thanks for the response. I have increased my score to 5.

---

### Official Review · Reviewer_rvWp · 2025-06-30

**Clarity:** 2
**Significance:** 2
**Originality:** 2
**Rating:** 4
**Confidence:** 5

**Summary:**

This paper studies the problem of preprocessing a Boolean $m \times n$ matrix $M$ to support fast matrix-vector product (MVP). The main result of this paper is that MVP can be supported in time $mn^{1 - 1/d}$ with $m n$ preprocessing time, where $d$ is the VC dimension of $M$.
As a bonus, this result extends to matrices that are "close" to the set $\mathcal N_d$ of matrices with VC dimension $\leq d$. Namely, $\min_{N \in \mathcal N_d} \max_i||M_i - N_i||_\infty \leq n^{1 - 1/d}$ suffice (denote with $A_i$ the $i$-th column of $A$).

Since MVP is a fundamental problem, the complexity bound above propagates to several other problems.

**Questions:**

You claim that your work bridges the gap between MVP heuristics used in practice and theoretical lower bounds. It would be interesting if you could further elaborate on why real-world MVP involve low-VC-dimensional matrices. Could you provide examples of successful heurstics which behavior is exampled by low VC dimension?
At line 83 you claim that "real-world data has low VC dimension": please elaborate more.

As written above, this work's results could be compressed into an extremely short note, and I believe that the community would benefit the most if this work was published (perhaps on Arxiv) as a coincise note.

**Ethical Concerns:**

["NO or VERY MINOR ethics concerns only"]

**Final Justification:**

I revise my evaluation and recommend acceptance, as long as there is room.

**Limitations:**

Yes.

**Paper Formatting Concerns:**

No.

**Quality:**

2

**Strengths And Weaknesses:**

This work is, to the best of my knowledge, the first to parameterize the complexity of MVP with respect to the VC dimension of a the matrix. The techniques employed to design the data structure achieving the desired time bound are the following.

First, the authors define a spanning tree for a matrix $M$ as a spanning tree of the weighted complete graph over $[n] \times [n]$, where $w(i, j) = ||M_i - M_j||_1$. Then, they prove a structural lemma showing that any matrix $M$ with VC dimension $d$ admits a spanning tree of weight at most $m n^{1-1/d}$. Finally, they observe that a spanning tree $T$ for $M$ can be leveraged to compute MVPs with $M$ in time $w(T)$.
To turn these observations into an algorithm, they use a known algorithm from Indyk and Motwani that computes a $O(\log n)$-approximate minimum spanning tree over hamming metric in  $\tilde O(mn)$ time.

This paper introduces a novel parameterization for the MVP problem and derives novel bounds. I believe that the main result of this work is interesting and certainly worth observing. Nonetheless, I recommend rejection due to the lack of technical novelty.

The technical novelty of this work is minimal. This work's contribution is a series of simple observation that I believe could be expressed in a four-page note, without compromising clarity.
Let me briefly evaluate the novelty (or lack thereof) of each of this work's main propositions.

Theorem 1.1 follows from the simple observation that each $n \times n$ matrix is the sub-matrix of any matrix with VC dimension at least $n$.

Lemma 3.4 can be proven through simple algebraic manipulations of the definition of MVP.

Lemma 3.12, which proves the main structural result of this work, follows by applying Lemma 4.1 of Welzl [1988] greedily.

Lemma 3.13 follows from the classic metric TSP analysis because crossing numbers are a metric.

Finally, and perhaps less importantly, the quality of writing is low. This paper could be much shorter, and thus more readable, if the authors did not add a long list containing several of the obvious consequences of a faster MVP algorithm. Moreover, the density of typos made it hard to understand at first sight. A list of writing mistakes and typos follows.

MISTAKES and TYPOS:
In the abstract, you do not say that your result holds for the class of Boolean matrices.

Line 78, "rows in S" is unclear. I guess you mean the sub-matrix which columns are restricted to those in S.

Line 79 - 84, you define VC dimension for matrices and then talk about hypothesis classes. A reader who's not familiare with learning theory would get lost here.

Line 90 - 91. I do not understand what you mean here.

Definition 1.2. I imagine that we're taking the minimum over such d.

Line 150 - 155. What is the threshold of a matrix?

Line 258. Regime -> paradigm.

Definition 3.8. I imagine the definition that you wanted to give is that A crosses r if $A \cap r, A \Delta r \neq \emptyset$

Line 368. \max -> \Sigma

---

> ### Author Rebuttal · Authors · 2025-07-30
>
> We thank the reviewer for their detailed comments. Firstly, we thank the reviewer for the careful reading and list of typos, and we will make sure to address these. In the answer below, we start by addressing comments on technical contributions. Then, we address items from the typo list that are not just small edits, or where we believe further context may be helpful.
>
> > Regarding technical contributions & 4 page note
>
> The reviewer’s comments refer to the proofs in section 3, which prove only the existence of low-weight trees. That part is indeed <4 pages long and (except for the Welzl citation) provides a self-contained proof. We felt that the narrative would be incomplete without formally proving these lemmas in Section 3, since we want to formally argue why the algorithm is efficient.
>
> We do provide many further technical contributions, such as an efficient algorithm for transposed matrices (which could have exponentially larger VC-dimension) as well as data structures for the dynamic case (where the matrix can change). The respective proofs are contained in the Appendix.
>
> > Theorem 1.1 follows from the simple observation that each nxn matrix is the sub-matrix of any matrix with VC dimension at least n.
>
> There seems to be a misunderstanding of what theorem 1.1 says. We have shown that for any non-trivial matrix family that is closed under row/column-deletion, all matrices in this family have a VC-dimension that is _at most_ a constant.
>
> - As a concrete example for such a matrix family, consider geometric matrices where rows and columns are assigned points in d-dimensional space and the 0/1 entries reflect distance < 1. For any such matrix, if we delete a row or column, then it is still such a geometric matrix. So this family is closed under deletion and thus every such matrix has constant VC-dimension (which is already known for this specific example).
>
> The point of Thm 1.1 is that we do not need to know what the structure is, e.g., it doesn’t have to be something that is defined via labels on rows/columns. We don’t need to know what exactly the specific structure/family is: as long as the family is closed under deletions, all matrices of that family have VC-dimension < some constant (i.e. independent of matrix size).
>
> - Our proof for this statement exploits several deep results from structural graph theory, so we respectfully disagree that it's a “simple observation”: If the reviewer thinks there is a simpler way to formally prove the theorem, we would definitely appreciate a clarification: at the moment, we don’t see any connection between Theorem 1.1 and the proposed simple observation.
>
> “Every $n\times n$ matrix A being a submatrix of a larger matrix B of VC-dimension at least n,” has no relation to Theorem 1.1 because (1) the larger matrix B is generally not part of the family, and (2) a _lower bound_ on B’s VC-dimension does not give a _constant upper bound_ on the VC-dimension of the submatrix A.
>
> > Results for Boolean matrices
>
> We will explicitly add “Boolean” to the abstract (it’s currently only implicit in the abstract since the VC-dimension is only defined for Boolean matrices). However, our results do extend to real/integer matrices (see Lines 150-155 and Appendix B.4) through our analysis with the Pollard pseudodimension.
>
> > Line 90-91: I do not understand what you mean here
>
> Theorem 1.1 specifies a sufficient but not necessary condition for constant VC-dimension. Lines 90-91 mean to say that matrices can have a constant VC-dimension without satisfying the condition of Thm 1.1 (i.e., there are also other conditions that imply constant VC-dimension. For instance, adjacency matrices of minor-free graphs have constant VC-dimension.)
>
> > Definition 1.2
>
> The corrupted VC-dimension of a matrix is the minimum such d. That being said, since we provide upper bounds on the time complexity, the parameter $d$ in our Theorems can be any upper bound on the corrupted VC-dimension. It just results in looser bounds.
>
> >Line 150 - 155. What is the threshold of a matrix
>
> The thresholds are part of the definition of Pollard Pseudodimension (an extension of VC-dimension to the non-Boolean setting). Line 153 should have referenced Appendix A where Pseudodimension is defined.
>
> When applied to real-valued matrix $\mathbf{M}$, the definition of Pseudodimension uses thresholds $T\in \mathbb{R}$ to construct Boolean matrices given by $\mathbf{M}_{i,j} \le T$. One never needs to consider more distinct thresholds than the number of distinct entries in M.
>
> > Definition 3.8. I imagine the definition that you wanted to give is that A crosses r if $A\cap r, A \Delta r \neq \emptyset$
>
> Thank you for catching this. Yes, the first formulation of the definition was wrong. The 2nd part (“$A$ crosses $r$ if neither $A \subseteq r$ nor $A\cap r=\emptyset$ holds”) is the correct definition, and that’s also the one we used in our proofs.
>
> The remaining listed typos will of course also be fixed. We thank the reviewer again for their valuable feedback. We hope our reply addresses any concerns in our work, and we would be happy to answer any further questions.
>
> Thanks,
>
> Authors

---

> > ### Comment · Reviewer_rvWp · 2025-08-01
> >
> > Thanks for your response. I Agree, Theorem 1.1 is *not* a consequence of the statement that I suggested.
> >
> > In your work, Theorem 1.1 plays the role of an observation, as it suggests that the "bounded VC dimension property" is natural and we should expect to hold in practice. On the other hand, it does not add much technical depth to your result.
> > All in all, I still believe that this work lacks in technical depth.

---

> > > ### Author Response · Authors · 2025-08-03
> > > **Response to Reviewer rvWp**
> > >
> > > > Thanks for your response. I Agree, Theorem 1.1 is not a consequence of the statement that I suggested. In your work, Theorem 1.1 plays the role of an observation, as it suggests that the "bounded VC dimension property" is natural and we should expect to hold in practice. On the other hand, it does not add much technical depth to your result. All in all, I still believe that this work lacks in technical depth.
> > >
> > >
> > > We thank the reviewer for their response.
> > >
> > > We agree with the reviewer on Theorem 1.1. It motivates why matrices with bounded VC-dimension are likely to occur in practice, but the theorem is not needed for proving our main results or applications.
> > >
> > > > Regarding the technical depth of this work:
> > >
> > > If the reviewer’s concern is the simplicity of the proof for fast matrix-vector products, we’d like to remark that this _should not detract from its novelty, but rather should improve its clarity and broader applicability_. Simplicity of a proof in hindsight does not mean the result is obvious. Despite the fundamental nature of matrix-vector multiplication as an algorithmic primitive and the existence of the used computational geometry techniques _for at least three decades_, these connections had not been previously recognized.
> > >
> > > That being said, our applications are not direct implications from just making these connections, but require developing further algorithms & data structures:
> > >
> > > 1) When the $\Delta$-labeled spanning tree is over the columns of the matrix (which is needed as VC-dimension of M and M-transposed can vary wildly), then the algorithm and its analysis is a lot more involved than for the row-based algorithms from prior work [1, 2]. Our correctness proof for the new column-based algorithm ends up being about 4 times longer than prior work’s row-based algorithm (Appendix B.1).
> > >
> > > 2) The applications require developing a dynamic variant (Theorem 1.4, where the rows and columns of the matrix can change) which is non-trivial, since existing data structures for maintaining (approximate) minimum spanning trees are not fast enough. Instead of maintaining a tree, we show that the dynamic setting can be solved by splitting the input matrix into $O(\log n)$ submatrices (Appendix B.3). This results in a reduction from dynamic to static matrices (Lemma B.4), so any other future faster MVP data structure for static matrices would also imply improvements for the dynamic setting.
> > >
> > >
> > > We hope this clarifies our perspective on the technical depth of our work, and we will make sure to better highlight the above two points.
> > >
> > >
> > > Thanks,
> > >
> > > Authors
> > >
> > > ===============
> > >
> > > References:
> > >
> > > 1) Fast boolean matrix multiplication for highly clustered data. Andreas Björklund and Andrzej Lingas. Workshop on Algorithms and Data Structures, 2001
> > > 2) Accelerating Graph Neural Networks Using a Novel Computation-Friendly Matrix Compression Format. Alves et. al. IEEE International Parallel and Distributed Processing Symposium (IPDPS) 2025

---

### Official Review · Reviewer_TRxg · 2025-07-02

**Clarity:** 3
**Significance:** 2
**Originality:** 4
**Rating:** 4
**Confidence:** 2

**Summary:**

The paper provides a theoretical explanation on the speed up observed in some recent practical algorithms for matrix-vector multiplication through preprocessing of matrices. In summary, the paper proves that for  an n × n matrix M of VC-dimension d, the matrix-vector product Mv can be computed in O(n^(2−1/d)) time for any vector v, after preprocessing in O(n^2) time. This is in contrast to the theoretical bounds showing that matrix-vector multiplication over sufficiently large fields require \Omega(n^2/ log n) time, where the authors claim that the speed up in practice comes thanks to the structure present, measured through the low VC dimension.

**Questions:**

- The authors argue that most real data has “low constant VC-dimensions”. Can you please better qualify this claim? Can you provide examples? A better example would be to show (in practice) an increase in the speed up with a decrease in the VC dimension of the data.

Minor issues:

- Line 248: semiring -> semi-ring

- Line 258: Can you better explain what you mean by “ algorithms with predictions regime” ?

- Line 261: “ Our dynamic algorithms” -> which algorithms does this refer to?

**Ethical Concerns:**

["NO or VERY MINOR ethics concerns only"]

**Final Justification:**

Thank you for your responses. I leave the decision of whether this is a good fit for the conference or not to the Area Chair. Based on all the reviews and the responses, I have increased my score to 4.

**Limitations:**

Yes. The work is purely theoretical.

**Paper Formatting Concerns:**

None.

**Quality:**

4

**Strengths And Weaknesses:**

Overall, this is an interesting theoretical paper that explains an important practical phenomenon. The paper is well written. It motivates the problem well with a very comprehensive background on the literature. This is far from my field of study, so I was not able to follow the proofs, but I was able to appreciate the solution steps and the logic behind the arguments.

On the other hand, the main weakness of the paper, in my opinion, is whether it is a good match for NeurIPS. I would have expected this result to appear in a more theoretically inclined venue.

From a more technical point of view: the solution mainly provides a (partial)  theoretical explanation of the speed up observed in practice, but does not provide any insights or guidance to come up with more efficient algorithms based on the VC dimension of data.

---

> ### Author Rebuttal · Authors · 2025-07-30
>
> We thank the reviewer for the kind review. Please find our response below.
>
> > Choice of venue
>
> The submission is indeed of a theoretical nature, and the NeurIPS call for papers explicitly lists theory as an area of interest. Given the central role of matrix-vector products in learning algorithms, and the prevalence of structured inputs in learning tasks, together with the fact that one of the analyzed algorithms was used to speed up learning in Graph Neural Networks [2], we believe that our results are a good fit for NeurIPS.
>
> > Technical point of view & Guidance/insights for algorithm design
>
> In addition to explaining the speed up of previous algorithms, we did also provide a new algorithm for when the matrix transpose has low VC-dimension (this is important since the previous algorithm is only efficient for the case where the matrix has low VC-dimension, but the transpose of a matrix can have a VC-dimension that is exponentially higher), as well as data structures for the dynamic setting (where the matrix is allowed to change over time).
>
> Regarding algorithm design, our results and new algorithms form a tool for constructing efficient VC-sensitive algorithms. Using our dynamic matrix-vector product algorithms, we provide several new algorithms for various dynamic graph problems in the applications section (e.g. dynamic effective-resistance and all-pairs-shortest-paths computation). For these problems, there are known $\Omega(n^2)$ lower bounds that hold for general graphs, but with our new VC-sensitive algorithms one can now construct faster $O(n^{2-1/d})$ upper bounds when the input is structured (i.e., has low VC-dimension).
>
> Notably, our data structures do not need to know the exact structure of the matrix/graph (e.g., the algorithms are not restricted to a specific structure like geometric intersection graphs), and they also do not need to know/compute the VC-dimension. Hence, our data structures are widely applicable as subroutines to other problems.
>
> Moreover, since computing the VC-dimension is $\Sigma_3^{\mathrm{P}}$-hard, no efficient algorithm should compute the VC-dimension. So if the dimension is needed by an algorithm, it would restrict applications to cases where the specific structure, and thus VC-dimension, is known.
>
> > Low constant VC-dimension in real data
>
> We thank the reviewer for this question. We would direct them to [1] which computes the VC-dimension for several real-world datasets (e.g. web graph, protein interaction networks, etc.) and shows they are low constants (i.e. between 3-8). Moreover, the matrix-vector product algorithm has been experimentally observed to be fast in practice in [2], but they haven’t studied the observed running time with respect to different VC-dimensions. We can add an experiment that “smoothly” parameterizes with d, and add a figure as well as an open-source code link for the implementation in the main body.
>
> > Question on algorithms with predictions regime
>
> “Algorithms with predictions” refers to algorithms where one has some prediction of the future/online input ahead of time. This model is studied in many different contexts (online algorithms, streaming, data structures, etc [5]). For matrix-vector product data structures, the assumption is that knowledge about the future vectors is known during preprocessing already, allowing to precompute information for future queries [3,4].
>
>
> > Question on “Our dynamic algorithms”
>
> Thanks for requesting more clarity on this. The algorithms we refer to are _all_ the dynamic algorithms in the paper, i.e. Theorems 1.6-1.10 (dynamic Laplacian solver, effective resistances, triangle detection, single-source shortest paths, $k$-center clustering), and theorem 1.4 which provides our dynamic algorithm for fast Mv.  To be clear, when we say “structured graph” in line 261, we mean a graph with constant VC-dimension. We will add clarifications for this in the paper.
>
> We thank the reviewer again for their valuable feedback. We hope our reply addresses any concerns in our work, and we would be happy to answer any further questions.
>
>
> Thanks,
>
> Authors
>
> ==============================
>
> References:
>
> 1) Practical Computation of Graph VC-Dimension. Coudert et. al. 2024. SEA 2024
> 2) Accelerating Graph Neural Networks with a Novel Matrix Compression Format. Alves et. al. arXiV 2024.
> 3) On Dynamic Graph Algorithms with Predictions. Van den Brand et. al. SODA 2024
> 4) On the complexity of algorithms with predictions for dynamic graph problems. Henzinger et. al. ITCS 2024
> 5) Algorithms with Predictions. Alexander Lindermayr and Nicole Megow. Survey webpage.

---

### Official Review · Reviewer_9Lbq · 2025-07-08

**Clarity:** 2
**Significance:** 3
**Originality:** 3
**Rating:** 5
**Confidence:** 3

**Summary:**

The paper considered the problem of preprocessing a $m\times n$ 0/1 matrix $M$, such that matrix vector queries on $M$ can be efficiently implemented. The main takeaway from this work is that if the VC-dimension of the rows of $M$ is at most $d$ then matrix vector queries can be implemented in time complexity $mn^{1-1/d}$. They have also considered both static and dynamic online variants of this problem. The paper as applications of the above result give efficient algorithms for the following problems

1. Dynamic Laplacian solver

2. Dynamic effective resistance of graphs

3. Dynamic triangle detection

4. Dynamic approximate single-source shortest paths

The authors fundamentally use ideas from computational geometry. Specifically, they use trees of low crossing number for set systems with low VC dimension, a fundamental result in both computational and combinatorial geometry.

**Questions:**

None

**Ethical Concerns:**

["NO or VERY MINOR ethics concerns only"]

**Final Justification:**

I am in favor of accepting this paper, as it makes a fundamental contribution to an important problem that is highly relevant to machine learning.

**Quality:**

3

**Strengths And Weaknesses:**

Strengths

1. The paper is well written and well motivated.

2. The overall presentation is good.

3. It introduces ideas from computational geometry to address a matrix problem in a novel way.

4. It gives multiple important applications of their central result.

There are no obvious weaknesses in this paper.

---

> ### Author Rebuttal · Authors · 2025-07-30
>
> We thank the reviewer for their kind review.
>
> Please don’t hesitate to follow up if any questions come up.
>
> Thanks,
>
> Authors

---

> > ### Comment · Reviewer_9Lbq · 2025-08-04
> >
> > Based on the current version of the manuscript, I remain in favor of accepting the paper for publication. I have just one minor suggestion: it would be helpful if the author(s) could make the dependence on $d$ more explicit in the constants hidden by the $O(\cdot)$ notation. This would address the important point raised by Reviewer goCC. While reading the manuscript, I had assumed that $d$ was a constant.

---

### Decision · Program_Chairs · 2025-09-17

**Decision:**

Accept (poster)

**Comment:**

Reviewers' scores are all positive (after rebuttal). The authors' response successfully addressed questions about some technical details. While some questioned the paper's fit for NeurIPS, given the wide use of matrix–vector multiplication in ML algorithms, I believe that this work fits NeurIPS well. One reviewer felt the technical contribution was somewhat insufficient, but the overall sentiment is that the technical contribution of this paper is above the NeurIPS acceptance bar.